

# Frequency Analysis of Extreme Sub-Daily Precipitation under Stationary and Non-Stationary Conditions across Two Contrasting Hydroclimatic Environments

Eleonora M.C. Demaria[1], David Goodrich[1], Timothy Keefer[1]

[1]Southwest Watershed Research Center, USDA-ARS, 2000 E Allen Rd, Tucson, 85719 AZ, USA

Correspondence to: Eleonora M.C. Demaria (eleonora.demaria@ars.usda.gov)

**Abstract.** Observed sub-daily precipitation intensities from contrasting hydroclimatic environments in the USA are used to evaluate temporal trends and to develop Intensity-Duration-Frequency (IDF) curves under stationary and non-stationary

climatic conditions. Analyses are based on observations from two United States Department of Agriculture (USDA)-Agricultural Research Service (ARS) experimental watersheds located in a semi-arid and a temperate environment. We use an Annual Maximum Series (AMS) and a Partial Duration Series (PDS) approach to identify temporal trends in maximum intensities for durations ranging from 5- to 1440-minutes. A Bayesian approach with Monte Carlo techniques is used to incorporate the effect of non-stationary climatic assumptions in the IDF curves. The results show increasing trends in

observed AMS sub-daily intensities in both watersheds whereas trends in the PDS observations are mostly positive in the semi-arid site and a mix of positive and negative in the temperate site. Stationary climate assumptions lead to much lower estimated sub-daily intensities than those under non-stationary assumptions with larger absolute differences found for shorter durations and smaller return periods. The risk of failure (R) of a hydraulic structure is increased for non-stationary effects over those of stationary effects, with absolute differences of 25% for a 100-year return period (T) and a project life (n) of 100

years. The study highlights the importance of considering non-stationarity, due to natural variability or to climate change, in storm design.

## 1 Introduction

Extreme rainfall events are linked to flooding, property damage, and potential loss of life. As the atmosphere gets warmer (Hartmann et al., 2013), changes in its water vapor holding capacity and shifts in circulation dynamics are expected to lead

to changes in the characteristics of precipitation. Positive trends in daily precipitation extremes have been reported globally and within the U.S. (Groisman et al., 2005; Kunkel et al., 2012; Mishra et al., 2012) with indications that sub-daily precipitation extremes are increasing beyond what is expected from the Clausius-Clapeyron relation (Lenderink and van Meijgaard, 2008; Westra et al., 2014). Climate models project substantial increases in water vapor concentration during the 21[st] century and it has been hypothesized that this will change the magnitude and frequency of intense precipitation events

(Emori, 2005; Janssen et al., 2014; Kunkel et al., 2013; Min et al., 2011). A contrast between positive trends in daily



precipitation extremes in the eastern U.S. and negative trends in the western U.S. has been documented (Kunkel et al., 1999; Pryor et al., 2009; Yu et al., 2016). There are indications that the southwest U.S. has become drier during the last several decades (Prein et al., 2016; Seager, 2007) and that in the arid and semi-arid regions located in subtropical zones an expansion of the sub-tropical highs (Seidel et al., 2008) and the poleward shift of the mid-latitude jet (Archer and Caldeira,

2008) are likely to increase heat and aridity in the future. In contrast, in the Northeast and Midwest regions observed increases in precipitation are projected to continue during the 21[st] century (Horton et al., 2014; Pryor et al., 2014).

Infrastructure design is commonly based on Intensity-Duration-Frequency (IDF) curves that are developed using rain gauge observations for different durations ranging from 5 minutes to 60 days. However, observations are not often available at sub-

daily time steps and design storms are estimated indirectly from daily totals using linear scaling factors as implemented in the National Oceanic and Atmospheric Administration (NOAA) Atlas 14 (Bonnin, G.M. et al., 2011). Although the intensification of precipitation events has been widely evaluated by the scientific community, the engineering community relies on the concept of return period (T) (also referred as average recurrence intervals or annual return period) to design hydraulic and storm water structures. An important assumption in IDF curves development is the stationarity of the model,

which implies that the probability of occurrence of events is not expected to significantly change in time. However, low-frequency components of climate variability such as El Nino Southern Oscillation (ENSO) and decadal and multi-decadal oscillations such as the Pacific Decadal Oscillation (PDO) and Atlantic Multi-decadal Oscillation (AMO) can introduce increasing or decreasing trends in the hydrologic records (Salas and Obeysekera, 2014). In addition, positive increases in observed precipitation linked to a warmer atmosphere also bring into question the validity of the IDF curves developed under

stationary assumptions and has prompted the development of new methods for determining return periods and risk that take into account the effect of non-stationary in climate extremes  (Bhatkoti et al., 2016; Condon et al., 2015; Du et al., 2015; Khaliq et al., 2006; Obeysekera and Salas, 2014; Salas and Obeysekera, 2014; Serinaldi and Kilsby, 2015). A few climate studies have shown that design storm estimates from stationary models are significantly lower than estimates from non-stationary models (Cheng and AghaKouchak, 2014; Sarhadi and Soulis, 2017; Wi et al., 2016). However, the lack of

observed sub-daily precipitation limits the studies to using scaling factors to temporally disaggregate daily observations or use merged radar-rain gauge data (Cheng and AghaKouchak, 2014; Yu et al., 2016). The United States Department of Agriculture (USDA)-Agricultural Research Service (ARS) network of experimental watersheds has been systematically measuring sub-daily precipitation since the mid-twentieth century and its database represents a unique opportunity to evaluate temporal changes and develop IDF curves in different hydroclimatic environments. The goals of this study are: (1)

to use observed sub-daily summer (June-October) precipitation intensities from two ARS sites, located in a semi-arid and a temperate climate, to test for evidence of temporal trends; (2) to build IDF curves using Annual Maximum Series (AMS) and Partial Duration Series (PDS) approaches and a Bayesian method that takes into account the non-stationarity of the time series; and (3) to estimate the risk of structural failure associated with design storms under stationary and non-stationary assumptions.



## 2 Data and methods

### 2.1 Study area and data used

Sub-daily precipitation intensities over the 1961-2014 period are selected for two ARS watersheds located in distinct hydroclimatic environments in the U.S.: the Walnut Gulch Experimental Watershed (WGEW) located in a semi-arid environment in Arizona and the North Appalachian Experimental Watershed (NAEW) located in a temperate environment in Ohio (Fig. 1 (a)). Table 1 shows the main characteristics of the watersheds. The semi-arid site is equipped with 56 rain gauges in order to capture the highly localized and scattered nature of summer convective storms in the southwestern United States which can have, on average, a spatial extent of 70 km$^2$ (Syed et al., 2003); the temperate site has available observations from 3 rain gauges. Both sites are equipped with Belfort 8-inch (0.2032 m) unshielded weighing-bucket rain gauges. Originally, a mechanical system recorded accumulated precipitation on paper charts and rainfall totals were later computed by manually digitizing the charts. In the mid-1990s, the analog-recording system was replaced by an electronic-weighing digital system that recorded rainfall with a 1/100-inch (0.254 mm) detection accuracy (Keefer et al., 2008; Owens et al., 2010). In order to build confidence in future analysis of the WGEW precipitation record, a direct comparison of event precipitation for nine pairs of co-located analog and digital rain gauges was undertaken for a 5 year period, 2000-2004 (Keefer et al., 2008). For an average of 260 events per pair, event total precipitation amount and peak intensities from 1 minute to 30 minutes were found to be equivalent. This analysis indicates that no artificial discontinuities were introduced by the switch in the recording system in a water-limited environment where highly localized, short duration, and high intensity summer storms are difficult to accurately observe. No comparable analysis has been carried out in the temperate watershed.

TABLE 1 HERE

The total seasonal precipitation (June-October) for the watersheds is presented in Fig. 1 (b) and (c). The semi-arid site shows large decadal variability with relatively drier seventies, wetter conditions in the eighties, and drier conditions in the beginning of the 21st century. There is a tendency towards drier conditions during the 54 years of record, however the trend is not statistically significant at a 1% significance level. Conversely, seasonal precipitation has steadily increased in the temperate site despite the strong natural variability that dominated the seventies (wet period) and the nineties (dry period). A statistically positive trend ($\alpha$ 0.1) of 2.47 mm/yr was found for the period (Fig. 1 (c)).

FIGURE 1 HERE

### 2.2 Time series and trend analysis

Two approaches were used to build IDF curves: 1) Annual Maximum Series (AMS) approach, also known as Block Maxima Approach, and 2) Partial Duration Series (PDS), also referred to as the Peak-over-Threshold approach. The AMS consists of





selecting the largest precipitation intensity for each year, thus the time series will have as many values as years in the record. This approach is widely used for extreme value analysis and for hydrologic design. The probability distribution of precipitation events selected with this method is well described by the Generalized Extreme Value (GEV) distribution function (Coles, 2001) given by:

$$F(x) = exp\left\{ -\left[ 1 + \xi\left(\tfrac{x-\mu}{\sigma}\right) \right]^{\tfrac{-1}{\xi}} \right\} \qquad (1)$$

where $u$ is the location parameter, $\sigma$ is the scale parameter and $\xi$ is the shape parameter under stationary conditions. Estimates of extreme quantiles $(z_T)$ used to estimate a design value or a T-year return level of the GEV distribution are given by:

$$z_T = \mu - \tfrac{\sigma}{\xi}\left[ 1 - \left\{ -log\left(1 - \tfrac{1}{T}\right) \right\}^{\xi} \right] \qquad (2)$$

The PDS is based on sampling all the observations that exceed an *a priori* specified threshold μ. Unlike the AMS approach, the PDS approach allows one to select more than one value per year and it provides additional flexibility to describe the natural variability of precipitation. Similarly to the AMS being described by GEV, the distribution of excess over the threshold y = x - μ has an approximate Generalized Pareto distribution (GPD) with the following cumulative distribution function (Katz et al., 2005):

$$F(y) = 1 - \left[ 1 + \xi\left(\tfrac{x-\mu}{\sigma^*}\right) \right] \qquad (3)$$

where ξ is the shape parameter, and σ* is the scale parameter which as for the GEV controls the spread of the distribution. The m-observational return level is defined as:

$$x_m = \mu + \tfrac{\sigma}{\xi}\left[ (m\zeta_\mu)^{\xi} - 1 \right] \qquad (4)$$

20    where ζμ = Pr(x>μ) and m is the number of observations above the threshold.

For AMS the return period is defined as the inverse of the exceedance probability (P) and it is easy to interpret since data are taken at fixed intervals. Thus the 1000-year flood has 0.001 chance, on average, to be exceeded in any given year. Conversely, for PDS, since multiple observations can be found in a year, the average number of observations (r) must be considered in the definition of T (Soong et al., 2004):

$$T = \tfrac{1}{rP} \; (in \; years) \qquad (5)$$

The PDS approach requires the selection of a high threshold to include extreme events in the analysis. Unfortunately, there is no reliable automated method to select the threshold in such a way that the independence criterion is satisfied and, at the same time, the sample size is equal to or larger than the AMS sample. Finding the right threshold is always a tradeoff between finding a value high enough to fit the GPD function to the data but not so high that too few observations are





selected which makes the statistical estimation of the distribution function parameters challenging (Coles, 2001). Conversely, a low threshold might lead to the selection of observations that are not statistically independent. The declustering methods available in the literature to meet the independence requirement make use of a fixed number of observations exceeding the threshold (Yilmaz et al., 2014), graphical approaches (Coles, 2001), or temporal windows

(Begueria, 2005). The reader is referred to Lang et al. (1999) for a review of available methods. In this study, we select the threshold based on three criteria: first the number of events exceeding the threshold to be larger than the number of years in the record; second the events are statistically independent; and third the goodness of the fitting of the GPD to the observed data. The GEV and GPD parameters were determined using the Maximum Likelihood (ML) method since likelihood-based techniques have shown better adaptability to the extreme value variation observed in natural systems (Coles, 2001; Hosking

and Wallis, 2005).

The presence of monotonic linear trends in precipitation extremes was evaluated with the Mann-Kendall (M-K) non-parametric test for each duration and for each rain gauge (Kendall, 1948; Mann, 1945; Sen, 1968). The magnitude of the trend was estimated with the Sen's method which has shown to be more robust since it is insensitive to outliers (Sen, 1968).

Trends were deemed statistically significant at a 0.1 significance level. Whenever the significance level is higher than the probability value (p) of the test statistic, the null hypothesis of no monotonic trend (trend absence) is rejected. The M-K test assumes independence (i.e. observations are not serially correlated over time). The presence of spatial (cross) correlation in precipitation data might affect the ability of statistical methods to assess the significance of a trend. When two rain gauges are highly correlated, the second one merely duplicates the information provided by the first one and might also lead to the

rejection of the null hypothesis (no trend) when in reality there was no trend. The Regional Average Mann-Kendall (RAMK) test incorporates the spatial (cross) correlation of the rain gauge network in the variance estimation (Yue and Wang, 2002). This study used the RAMK test to evaluate trends in AMS time series in both sites.

### 2.3 Non-stationary GEV and GPD estimation

When time series of AMS or PDS are known to be changing in time, time variant parameters should be fitted to the observations. The commonly used approach to incorporate non-stationarity in time series into the frequency analysis is to vary one parameter of the theoretical probability distribution while the other two are kept constant (Liuzzo and Freni, 2015; Wi et al., 2016). In this study we used a Bayesian approach to estimate distribution parameters under stationary and non-stationary assumptions (Cheng and AghaKouchak, 2014). A Monte Carlo sampling approach is implemented to infer the

posterior parameter distribution which allows uncertainty bounds to be obtained for estimated precipitation intensity values for different return periods. Parameters were estimated with the ML method and the goodness of the model fit was assessed with the $\hat{R}$ criterion (Gelman and Shirley, 2011). An initial set of 50,000 random samples were used to estimate the GEV and GPD posterior parameter distributions for each duration. From these, 10,000 samples were not included in the analysis





(burn-in) which reduced the final number of parameters to 40,000. The observed non-stationarity in precipitation extremes was modeled with a time-variant linear model of the location parameter $\mu(t) = \mu_1(t) + \mu_o$ in the GEV. For the GPD model, the threshold parameter was assumed to be stationary and only a linear time-variant model of the scale parameter ($\xi$) was used. The observed trends were used as a proxy for future trend magnitudes and utilized to compute the return levels as the

median of the 40,000 parameter samples multiplied by the linear trend over a 100-year period into the future. This approach implicitly assumes that the magnitude of the observed trend will remain constant in the future.

To measure the goodness of the fitting of observations to the theoretical distribution functions we used quantile-quantile (QQ) plots (Katz et al., 2002). To this end, the Weibull plotting positions were used to compute the empirical quantiles and

the theoretical quantiles were obtained from the distributions fitted to observed precipitation intensities. This method was only implemented for the stationary models since the parameters of the non-stationary models change with time. To prove that the non-stationary extreme value distributions appropriately described the observed precipitation intensities, both stationary and non-stationary models were tested with the deviance statistic (D). Considering the stationary model $M_0$ as a restrictive case of non-stationary model $M_1$ where $\mu_1 = 0$ for the GEV model and $\xi_1 = 0$ for the GPD model, it is possible to

test the null hypothesis of $M_0$ fitting to the data better than $M_1$ with D which is defined as:

$$D = 2\{l_1(M_1) - l_0(M_0)\} \hspace{3cm} (7)$$

where $l_l M_1$ and $l_0 M_0$ are the maximized log-likelihoods of models $M_1$ and $M_0$, respectively. Large D values indicate that model $M_1$ explains more of the variation in the data hence the increase in the model size improves its skill to explain

observed precipitation intensities. The validity of model $M_0$ relative to $M_1$ is tested at the $\alpha$-level if D > $c_\alpha$, where $c_\alpha$ is the (1-$\alpha$) quantile of a $\chi_k^2$ distribution and k is the difference in the dimensionality between the stationary and non-stationary models (Coles, 2001; Delgado et al., 2014).

The traditional risk analysis used for hydrologic design uses the concept of return period (i.e. the inverse of the exceedance

probability for a given magnitude) which is assumed to be constant in time during the period of interest. However, when observations show a temporal trend the exceedance probability varies with time and it can be used to extend the concept of risk analysis used in stationary risk design to non-stationary conditions. Non-stationary return periods based on time varying exceedance probability are computed using waiting time theory which incorporates the upward trend in observations. The hydrologic risk associated with stationary and time-varying precipitation intensities can be computed as:




$$R = 1 - \left(1 - \frac{1}{T}\right)^n \quad or \quad R = 1 - \prod_{t=1}^{n}\left(1 - \frac{1}{T_t}\right) \tag{8}$$

where n is the length of time of the project life, T is the stationary return period, and $T_t$ is the corresponding non-stationary return period.

## 3 Results

### 3.1 Trend analysis of precipitation extremes

For the analysis of temporal trends, 56 rain gauges were used in the semi-arid watershed and 3 rain gauges in the temperate watershed. Maximum rainfall intensities for each day were obtained for short: 5-, 15-, 30-minute, intermediate: 60-minute, and long: 1440-(1 day) minute durations for the warm months (June-October). We generated a time series of AMS at each gauge by selecting the maximum intensity for each year and each duration. The time series of PDS intensities were obtained by selecting the 95[th] percentile for all maximum precipitation intensities, this percentile value (assumed constant in a non-stationary climate) allowed an average sample size of 1.6 events per year. Previous studies have used percentiles ranging from the 90[th] to the 98[th] to define what constitutes an extreme event (Begueria et al., 2011; Groisman et al., 2005; Villarini et al., 2013). A 24-hour window was imposed between two events to ensure the statistical independence of PDS intensities. Outliers in the observations were removed using the Adjusted Boxplot method (Hubert and Vandervieren, 2008). The number of outliers was, on average 2($\pm$ 3.1) and 10($\pm$ 2.0) for all the rain gauges for the semi-arid and temperate sites, respectively. The effect of removing outliers on the trend analysis was negligible (not shown). Table 2 shows the number of rain gauges with positive(negative) trends, both statistically and not statistically significant, in precipitation intensities for the semi-arid(temperate) site. Both sites show a mix of positive and negative trends for AMS and PDS intensities for all durations. Overall, the semi-arid watershed shows positive trends in 72 %(68 %) of the gauges for AMS(PDS). However, the number of rain gauges with statistically significant trends is lower at 11 %(15 %), respectively. The regional trend analysis with RAMK shows that the 5- and 1440-min durations are statistically significant. Note that RAMK only allows computation of the trend for the AMS since it requires time series of the same length. In the temperate watershed, positive trends in AMS maximum intensities are statistically significant for 15-, 30-, and 1440-minute durations whereas PDS maximum intensities are mostly negative for short and intermediate durations. The RAMK test indicates that trends are statistically positive for 15-, 30-, and 1440-minute durations. The lack of a clear signal in the direction of the trends has been previously reported for other regions (Sarhadi and Soulis, 2017) and might be linked to the weak power of statistical methods when the data are affected by high natural variability as is the case of precipitation (Renard et al., 2008).

From the available rain gauges in each watershed, we selected one gauge to perform the IDF analysis for all the durations. Dimensionless trends were also computed as the magnitude of the trend in mm/hr, estimated with the Sen's method, divided





by the maximum intensity. These trends enable comparison of their magnitudes between the different durations. Figure 2 shows linear trends in AMS and PDS precipitation intensities. Positive trends ranged from 5.01 mm/hr per decade for the 5-min durations to 0.11 mm/hr for the 1440-min durations in the semi-arid site. Even though the magnitude of the trends declined with increasing durations, the smaller dimensionless trends were found for the 30- and 60-min durations and the

largest values for the 1440-min duration. In the temperate site, trends ranged from 3.21 to 0.19 mm/hr per decade for between the 5-min and 1440-min durations, however the smallest(largest) dimensionless trend corresponded to the 15-min(1440-min) duration. For the PDS observations, the trend analysis in the semi-arid site revealed increasing statistically significant trends for durations 15-, 30- and 1440-min. A noticeable difference between the AMS and PDS trend magnitudes at the temperate site is the identification of negative trends in PDS extremes. The decreasing trends in precipitation extremes

are likely related to the natural variability of precipitation with wetter-than-normal conditions in the beginning of the record.

       FIGURE 2 HERE

### 3.2 Extreme analysis of maximum precipitation intensities

Figure 3 shows the fitting of GEV and GPD models to precipitation intensities in the semi-arid ((a) and (b)) and temperate ((c) and (d)) watersheds, respectively. The fitting of the theoretical distribution functions under stationary conditions was found to be statistically significant in all rain gauges (5 % significance level) with the Anderson-Darling (A-D) and Kolmogorov-Smirnov (K-S) non-parametric tests. The GEV distribution, fitted to observations with the ML method, is able to appropriately represent the distribution of annual maximum intensities for the different durations. The ML method was

selected for the analysis since it easily adapts to different models including non-stationary models (Coles, 2001). The PDS extremes in both watersheds were also tested for the GPD with the ML estimator (Fig. 3 (b) and (d)). The results of the A-D and K-S tests suggested that this distribution can properly model the observed distribution of extremes over the 95[th] percentile in the different hydroclimatic environments.

FIGURE 3 HERE

       A visual evaluation of the fit of the GEV or GPD distributions to the time series of precipitation intensities is provided by the QQ plots (Fig. 4). For the GEV distribution based on Eq. (1) the QQ plot indicates that the fit is reasonably adequate although there is strong evidence of a heavy tail for large intensities in both sites that the theoretical distribution might not be

able to represent (Fig. 4 (a) through (e)). When the PDS approach is used, the fit of the GPD model to observations based on Eq. (3) appears to be a better model to represent time series of maximum intensities in the semi-arid site (Fig. 4 (f) through (j)). Note that the temperate site is not included in the analysis since no statistically significant trends were identified for that site. For all 40,000 Monte Carlo realizations, we obtained the maximized log-likelihoods for the stationary and non-





stationary models to compute the deviance statistic D (Eq. (10)). The significance level ($c_\alpha$), obtained from the $\chi^2_k$ distribution for α= 0.05, is equal to 3.84. Large D values indicate that the non-stationary model explains more of the variation in precipitation intensities than the stationary model (Table 3). In the semi-arid site, small values of D for all durations indicate that the non-stationary GEV model, with a linear trend in the location parameter, might not be a

significant improvement over the stationary version of the model despite the statistically significant positive trends found in Section 3.1. The D values for the temperate site exceed $c_\alpha$ for the 5-, 30-, and 1440-min supporting the incorporation of a linear trend in μ in the GEV models. There is strong evidence, D values exceeding the significance level, to support a non-stationary GPD model with a time-variant scale parameter for the semi-arid site.

FIGURE 4 and Table 3 HERE

### 3.3 Design storms and risk under stationary and non-stationary conditions

In this section, we address the issue of how design storms differ under stationary and non-stationary assumptions? An
answer to this question is relevant since it might inform practitioners about the uncertainty in the estimates when designing storm water infrastructure. In Figure 2, by examining storm durations, we found that a subset of sub-daily intensities show statistically significant increasing linear trends. With this in mind, a new set of IDF curves were developed with a framework that incorporates temporal changes in precipitation intensities. For both extreme value models, the magnitudes of the trends observed in the study period are assumed to continue at the same rate in the next 100 years into the future. Estimated
precipitation intensities (return levels) for different return periods, and the corresponding credibility interval (this term is called confidence interval in Frequentist context (Renard et al., 2008)) are shown in Fig. 5. In the semi-arid and temperate sites, the GEV estimated intensities under stationary climatic conditions are much lower than those under the non-stationary assumptions (Fig. 5 (a) to (j)), i.e., extreme precipitation intensities are likely to occur more frequently if time-variant conditions are incorporated in the analysis. Note that durations with no statistically significant trends were excluded from
the analysis, therefore both curves fall on top of each other in the plot. Conversely, the GPD model does not show appreciable difference between both climatic conditions in the semi-arid watershed. This could be the result of less steep trends (compared to the AMS data) despite the larger D values found (Table 3). These results indicate that a specific precipitation intensity will be more likely to occur, i.e., shorter return period, in the future if the observed trends are incorporated in the IDF design.

FIGURE 5 HERE



The IDF curves for each site are shown in Fig. 6. The largest absolute differences in GEV-intensities are found for short durations (changes are computed as the absolute difference between non-stationary and stationary) at both sites. For example, for the semi-arid site the estimated 2-year 5-min storm (5-minute duration with a 2-year return period) is 99.3 mm/hr under stationary assumptions and 122.3 mm/hr under non-stationary conditions, which represents an absolute

difference of 22.9 mm/hr, while for the estimated 100-year 5-min storm estimates are 189(204) mm/hr for stationary(non-stationary) which represents an absolute change of 15.5 mm/hr. For longer event durations, the absolute differences between stationary and nonstationary become smaller in absolute terms, ranging from 0.75 mm/hr to 0.59 mm/hr for the 2-year and 100-year return periods, respectively. In the semi-arid site, differences in GPD-estimated intensities are almost negligible. These results indicate that larger changes in design storms are likely to happen for shorter durations and smaller return

periods under warmer-than-normal climate conditions in agreement with previously found evidence that hourly and sub-hourly precipitation intensity is more sensitive to atmospheric temperature changes see Westra et al. (2014) and the references therein. This demonstrates that the stationary framework currently used for structural design systematically underestimate short-duration precipitation extremes which might lead to more frequent infrastructure damage.

FIGURE 6 HERE

We have shown in the previous sections that there is evidence of increasing precipitation intensities for certain durations during the period 1961-2014 (54 years). This implies that the exceedance probability of a certain event occurring in a given year will also vary with time and hence the risk of failure (R) for a specific T and project life ($n$) will change accordingly.

The difference in R for the semi-arid and temperate sites is shown in Fig. 7 for the 15- and 1440-min duration AMS precipitation intensities, for T= 10, 50, and 100-yr under stationary conditions and the corresponding $T_t$ under non-stationary assumptions. The insets in each panel show that Tt is systematically smaller than T at both sites and for the two selected durations. For instance, for the 15-min duration intensities the Tt value for a 50-year return period is 36(31) years for the semi-arid(temperate) whereas a Tt of 46 years corresponds to T= 100-year in both sites. Under stationary conditions, R

increases in accordance with the project life durations for the three return periods. However, the risk of failure is larger under the non-stationary conditions as the project life increases. For example, in the semi-arid site for a project life $n$=50 years and a return period T=10 years ($T_t$= 8.8 years) the absolute differences between stationary and non-stationary conditions are negligible. Conversely, for $n$=100 years and a return period T=100 years ($T_t$ = 46 years) the risks are 63%(89%) for stationary(non-stationary) conditions in the semi-arid watershed. This represents a difference in risk levels of 25%. Note that

we are using absolute differences no relative differences in this analysis. The risk of failure is very similar for the temperate site for smaller $n$, for a larger project life the absolute difference decreases to 18% for T=100 years (Fig. 7 (b)). The shape of the curves for the different durations differ between sites and no clear pattern between durations and the magnitude of R is found. This behavior can be a function of the parameters of the GEV distribution fitted to the observations and the





magnitude of the observed trend. However, despite these differences, the shape of the curves and the magnitude of the R are comparable for both sites.

FIGURE 7 HERE

The magnitude range of R for all the considered project life values ($n$ =1 to 100 years) and different return periods is shown Fig. 8. As expected, the probability of precipitation intensities exceeding the given design storm during the n-year of the project decreases for larger return periods since more intense precipitation intensities are less frequently observed. Changes in the median R (i.e., differences between non-stationary and stationary conditions) increase from 0.25 % for T=10 years to

27 %(18 %) for T = 100 years in the semi-arid(temperate) site. The uncertainty, reflected in the inter-quantile distance (25th to 75th percentiles), grows as the return period increases. The temperate watershed shows slightly larger differences between stationary and non-stationary durations, however the magnitudes are comparable for both sites. There is a close agreement between the magnitude of non-stationary R and the magnitude of the dimensionless trend shown in Fig. 2. These results indicate that despite the differences in precipitation intensities between stationary and non-stationary conditions are larger

for shorter storm durations (Fig. 5), more intense-short duration events under non-stationary climate conditions are not necessarily linked to a larger risk of failure perhaps because these differences are not large enough to be captured by Eq. (8).

FIGURE 8 HERE

When R is plotted over a range of precipitation intensities for the semi-arid site (Fig. 9), as expected R increases as a power function of the project life (Eq. (8)) and decreases with precipitation intensity (large intensities correspond to larger T or less frequent events). Note that precipitation intensity values are computed with the fitted GEV for set return period values. For the smaller $n$ (10-year), the differences in the median of R between stationary and non-stationary conditions do not exceed 10 % for the different event durations. As $n$ becomes larger, the differences increase up to 27 % for all durations. Overall, the

increased risk from the stationary condition to the non-stationary case is between 10 to 27 % for precipitation intensities ranging from 5 to 200 mm/hr as the project life grows.

FIGURE 9 HERE

**4 Summary and conclusions**

Engineering planning and water infrastructure design rely on IDF curves that have been developed under the assumption that climate conditions are stationary. This study shows that in two contrasting hydroclimatic environments the changing nature





of precipitation intensities over time leads to the underestimation of extreme storms that are used for hydrologic design. Our analysis is unique since it incorporates observed precipitation intensities at sub-daily durations and non-stationarity into the design IDF curves. Sub-daily precipitation intensities (period 1961-2014) at two USDA-ARS watersheds located in the United States: the semi-arid climate of Arizona and the temperate climate of Ohio are used to generate Annual Maximum

Series (AMS) and Partial Duration Series (PDS) for durations ranging from 5- to 1440-minutes. For each duration, first the presence of temporal linear trends is assessed with a non-parametric test and second, in the cases of statistically significant trends, IDF curves and the hydrologic risk of failure under stationary and non-stationary assumptions are computed using a Bayesian approach that incorporates time-variant parameters.

We find increasing trends in observed AMS sub-daily intensities in both sites. The PDS observations show positive trends in the semi-arid site, and a mix of positive and negative not statistically significant trends in the temperate site, perhaps due to long-term variability. Trends in AMS(PDS) observations are incorporated in a GEV(GPD) model with time-variant parameters expressed as a linear function of time to estimate precipitation intensities. In general, non-stationary models have improved skills in explaining the variability of precipitation intensities in both watersheds.

Using the derived GEV and GPD stationary and non-stationary models, we generate IDF curves for the different event durations and return periods. Despite the hydroclimatic differences between the semi-arid and the temperate site, the absolute differences in GEV-estimated precipitation intensities between stationary and non-stationary conditions are comparable for both sites. Estimated sub-daily intensities under stationary climatic conditions are much lower than those under the non-

stationary assumptions. For instance, for the semi-arid site the absolute difference in the 2-year 5-min storm is 22.9 mm/hr while for the 100-year 5-min storm the difference was 15.5 mm/hr. For longer event durations, the absolute difference become smaller. These changes in design storms under non-stationary conditions might lead to under-design of critical infrastructure not able to withstand future storms occuring more frequently than planned.

The risk of failure for non-stationarity climate conditions, due to natural variability or to climate change, exceeds that for stationary assumptions at both sites. The differences in risk under both climate conditions increase as a function of $n$ (project life), independently of the event duration, and for large return periods. The absolute difference in risk levels for $n$=50 years and T= 10 years is almost negligible, conversely the difference is 25 % for $n$=100 years and T= 100 years. These results indicate that the increase in precipitation risk under non-stationary climate conditions might be considered for projects

requiring a greater level of protection.

The results are valid for the two sites included in the study; however, the methodology can be implemented in different hydroclimatic environments where sub-daily precipitation observations are available to update IDF curves used for infrastructure design. However, our approach has one caveat. We base the analysis on the assumption that historical trends in





precipitation intensities are expected to continue at the same rate in the future. Even though we agree that this is not necessarily expected and that the projected trend magnitude, and its uncertainty, can be estimated from climate model projections, we argue that the historic work provides a baseline for the magnitude of trends in precipitation intensities. Our results can be used as a basis to incorporate non-stationarity in the design storm process and could be further updated with
more observations or as climate model efforts acquire more skill, and would move the design community forward where design life could be coupled with intensity trend projections to obtain more realistic design storms.

**Data and code availability**

The precipitation data can be found at http://www.tucson.ars.ag.gov/dap/ and the Matlab codes used for the analysis are available upon request from the corresponding author.

**Competing interests**

The authors declare that they have no conflict of interest.

**Acknowledgments.** We are thankful to the many USDA-ARS employees whose effort and dedication throughout the years have made the precipitation database available.

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





## TABLES

**Table 1**. Selected watershed for the extreme event analysis

| Experimental Watershed | State | Lat. | Lon. | Elev. [m] | Annual Prec. [mm] | Temp [C] | Rain gauges | Area [km²] | Climate Characteristics |
|---|---|---|---|---|---|---|---|---|---|
| Walnut Gulch | AZ | 31.72 | -110.01 | 1,200 | 312 | 17.7 | 56 | 149 | Semi-arid climate with high-intensity summer convective precipitation |
| North Appalachian | OH | 40.37 | -81.8 | 359 | 944 | 9.7 | 3 | 4.2 | Temperate Continental climate with convective thunderstorms in late spring and summer |

**Table 2**. Number of rain gauges showing positive/negative and statistically significant positive/negative trends in
10 precipitation intensities for different durations. Totals are computed as percentage of the number of rain gauges in each
watershed. NA (not applicable) indicates that the method was not used for these time series.

| | | | Positive | | Significant Positive | | Negative | | Significant Negative | | RAMK | |
|---|---|---|---|---|---|---|---|---|---|---|---|---|
| | **Duration (min)** | | AMS | PDS | AMS | PDS | AMS | PDS | AMS | PDS | AMS | PDS |
| semi-arid | short | 5 | 41 | 32 | 9 | 10 | 15 | 19 | 0 | 6 | SS | NA |
| | | 15 | 38 | 37 | 5 | 6 | 20 | 21 | 0 | 1 | - | NA |
| | | 30 | 40 | 41 | 2 | 6 | 17 | 17 | 1 | 1 | - | NA |
| | intermediate | 60 | 41 | 38 | 5 | 7 | 17 | 20 | 1 | 1 | - | NA |
| | long | 1440 | 50 | 48 | 12 | 14 | 6 | 7 | 0 | 1 | SS | NA |
| | Total [%] | | 72 | 68 | 11 | 15 | 26 | 29 | 1 | 3 | | |
| temperate | short | 5 | 2 | 1 | 0 | 0 | 0 | 2 | 0 | 1 | - | NA |
| | | 15 | 3 | 0 | 3 | 0 | 0 | 1 | 0 | 0 | SS | NA |
| | | 30 | 3 | 1 | 2 | 0 | 0 | 2 | 0 | 0 | SS | NA |
| | intermediate | 60 | 3 | 0 | 0 | 0 | 0 | 3 | 0 | 3 | - | NA |
| | long | 1440 | 3 | 3 | 3 | 1 | 0 | 0 | 0 | 0 | SS | NA |
| | Total [%] | | 93 | 33 | 53 | 7 | 0 | 53 | 0 | 27 | | |





**Table 3**. Deviance statistic (D) between stationary and non-stationary models. The significance level of D is computed with a $\chi^2_k$ distribution ($\alpha = 0.05$). Values of D larger than the 3.84 significance level are denoted in bold type.

| | | 5-min | 15-min | 30-min | 60-min | 1440-min |
|---|---|---|---|---|---|---|
| GEV | semi-arid | 0.5 | 2.2 | 0.4 | 1.4 | 3.4 |
| | temperate | **9.5** | 3.3 | **4.6** | 1.2 | **6.7** |
| GPD | semi-arid | **8.5** | **4.0** | **6.6** | **5.2** | 1.5 |
| | temperature | - | - | - | - | - |




**FIGURE CAPTIONS**

**Figure 1**. Geographic location of the study sites (a). Total seasonal precipitation for the warm months (June-October) for the period 1961-2014 for: (b) the semi-arid site (Walnut Gulch Experimental Watershed) and (c) the temperate site (North Appalachian Experimental Watershed). Thin gray lines denote a rain gauge, the blue line denotes the watershed-averaged

values, the black line denotes the 11-year watershed-averaged running mean, and the dotted blue line shows the linear trend.

**Figure 2**. Linear trends in AMS and PDS time series for two selected rain gauges. The numbers in each plot indicate: the magnitude of the slope per decade, and dimensionless trends defined as the magnitude of the slope divided by the largest precipitation intensity. Bold numbers denote statistically significant (alpha 0.1) trends.

**Figure 3**. Theoretical models for precipitation intensities in: semi-arid watershed top panels and temperate watershed bottom

panels for 5-, 15-, 30-, 60-, and 1440-min durations. GEV (a) and (c) and; GPD (b) and (d), respectively.

**Figure 4**. Model diagnosis using quantile-quantile (QQ) plots for stationary models: GEV top panels, GPD bottom panels.

**Figure 5**. Return periods vs. precipitation intensity return levels under stationary and non-stationary assumptions. The top two panels for GEV model: plots a. to e. semi-arid site, and plots f. to j temperate site. The bottom panels for the GPD model: plots k. to o. semi-arid site. The shaded band represents the $5^{th}$-$95^{th}$ percentiles of the posterior distribution

(credibility interval). The solid orange/green lines denote the median of the return level posterior distribution.

**Figure 6**. Stationary and non-stationary IDF curves for different return periods and durations at the semi-arid (Arizona) and the temperate (Ohio) watersheds. Plots in the right most panels (p. to r.) show changes between estimated non-stationary and stationary intensities as percentage of the stationary estimates (delta= $(X_{non-sta}-X_{sta})/X_{sta} * 100$).

**Figure 7**. Risk of failure as a function of project life (years) assuming stationary return periods (solid lines) T= 10, 50, and

100-year and the corresponding non-stationary Tt return periods (symbols). a. semi-arid site and b. temperate site. In both cases the 15- and 1440-min duration intensities are represented with the GEV model. The subplot inset shows the variation of Tt as a function of T.

**Figure 8**. Probability of precipitation (risk) for different project life (n: 1 to 100-year) versus return periods in years. Precipitation intensities for durations ranging from 5- to 1440-min are represented with the GEV model. Orange(green) lines

show stationary(non-stationary) conditions. a) semi-arid site and b) temperate site. The notch extremes correspond to the 25th and 75th percentiles, respectively, and the extent of the box correspond to the maximum and minimum values.

**Figure 9**. Risk of failure as a function of precipitation intensity in the semi-arid site for four project life: a) 10-year, b) 50-year, and c) 75-year and d) 100-year under stationary (orange) and non-stationary (green) conditions. Risk is computed for 1- to 100-yr return periods. Precipitation intensities are represented with the GEV model.








**FIGURES**

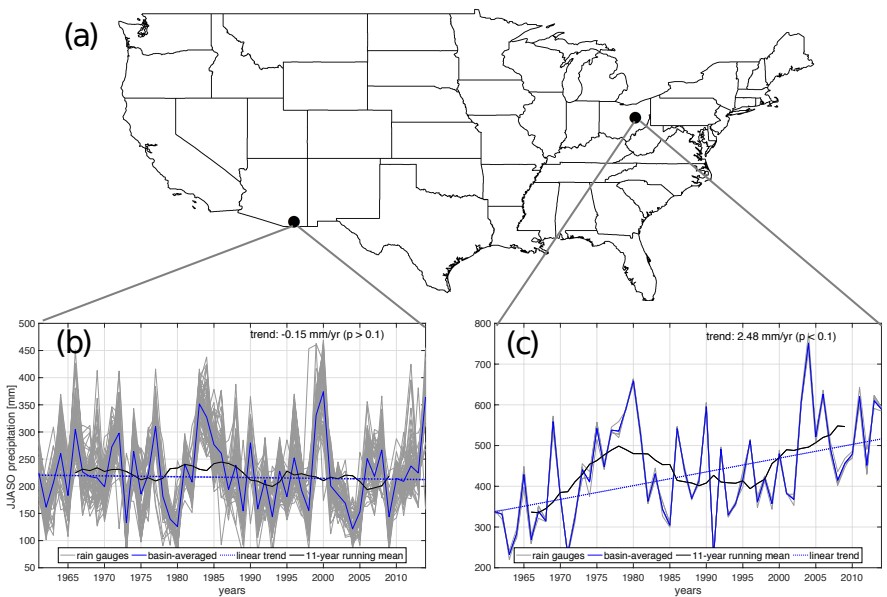

**Figure 1**. Geographic location of the study sites (a). Total seasonal precipitation for the warm months (June-October) for the
period 1961-2014 for: (b) the semi-arid site (Walnut Gulch Experimental Watershed) and (c) the temperate site (North
Appalachian Experimental Watershed). Thin gray lines denote a rain gauge, the blue line denotes the watershed-averaged
values, the black line denotes the 11-year watershed-averaged running mean, and the dotted blue line shows the linear trend.





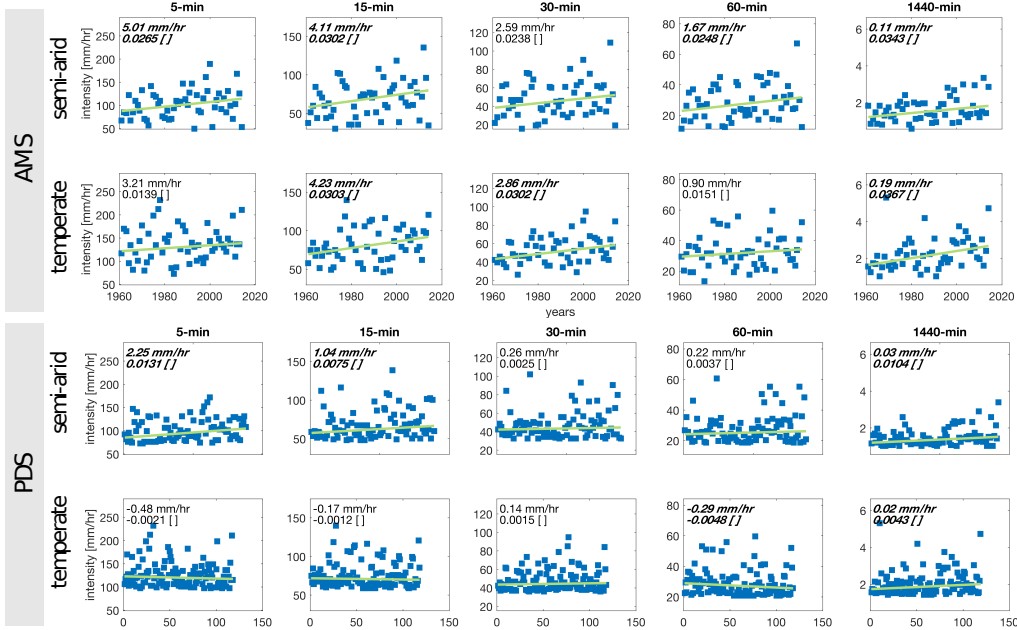

**Figure 2**. Linear trends in AMS and PDS time series for two selected rain gauges. The numbers in each plot indicate: the magnitude of the slope per decade, and dimensionless trends defined as the magnitude of the slope divided by the largest precipitation intensity. Bold numbers denote statistically significant (alpha 0.1) trends.





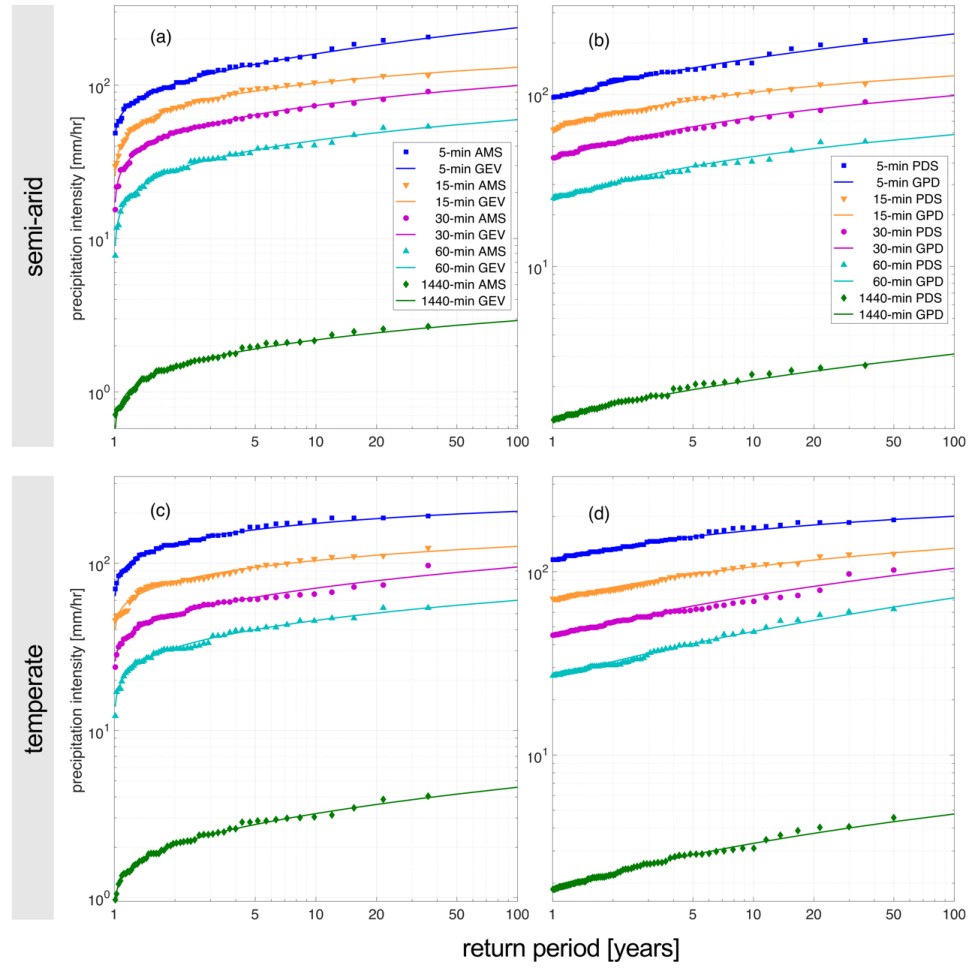

**Figure 3**. Theoretical models for precipitation intensities in: semi-arid watershed top panels and, temperate watershed bottom panels for 5-, 15-, 30-, 60-, and 1440-min durations. GEV (a) and (c) and; GPD (b) and (d), respectively.





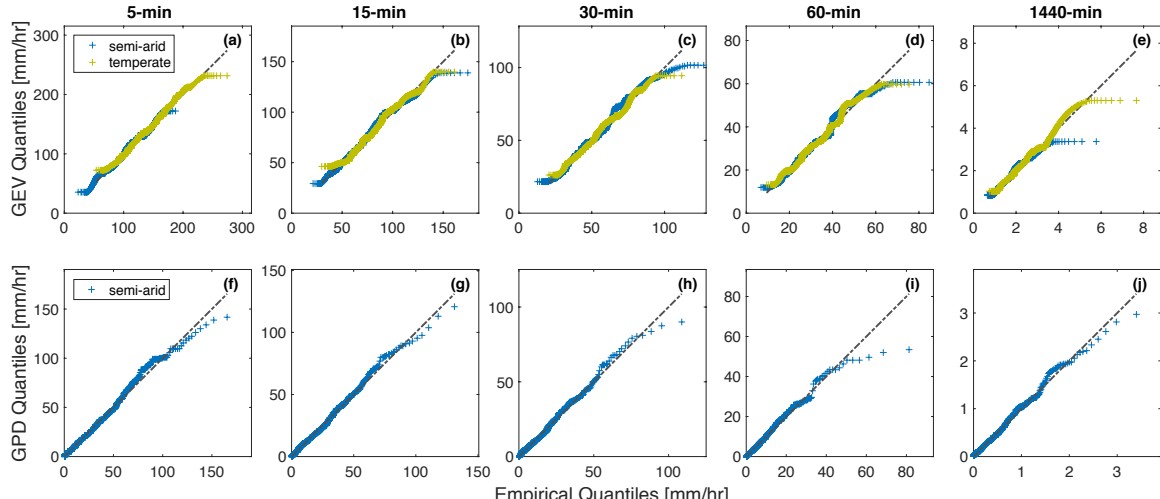

**Figure 4**. Model diagnosis using quantile-quantile (QQ) plots for stationary models: GEV top panels, GPD bottom panels.





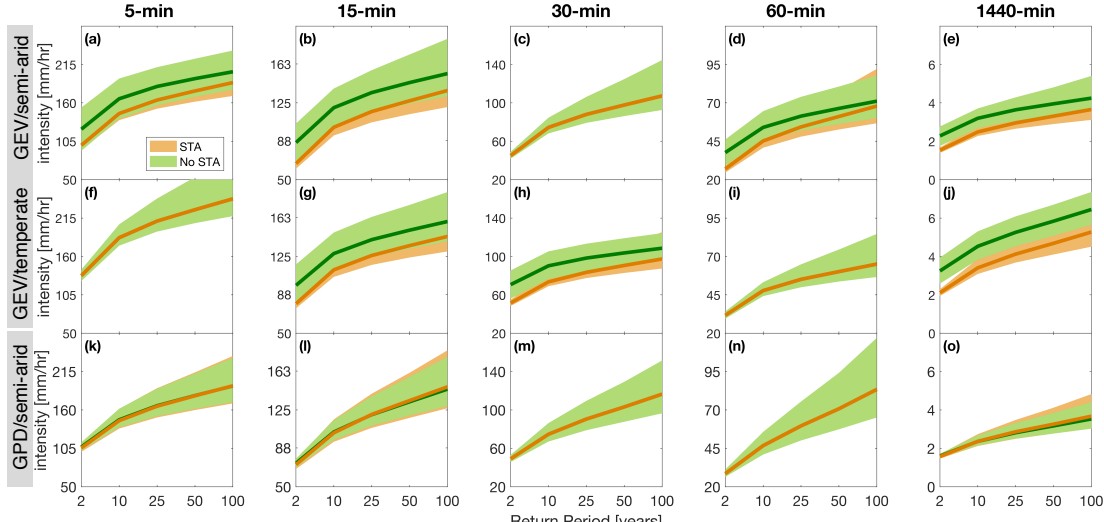

**Figure 5**. Return periods vs. precipitation intensity return levels under stationary and non-stationary assumptions. The top two panels for GEV model: plots a. to e. semi-arid site, and plots f. to j temperate site. The bottom panels for the GPD model: plots k. to o. semi-arid site. The shaded band represent the 5th-95th percentiles of the posterior distribution (credibility interval). The solid orange/green lines denote the median of the return level posterior distribution.





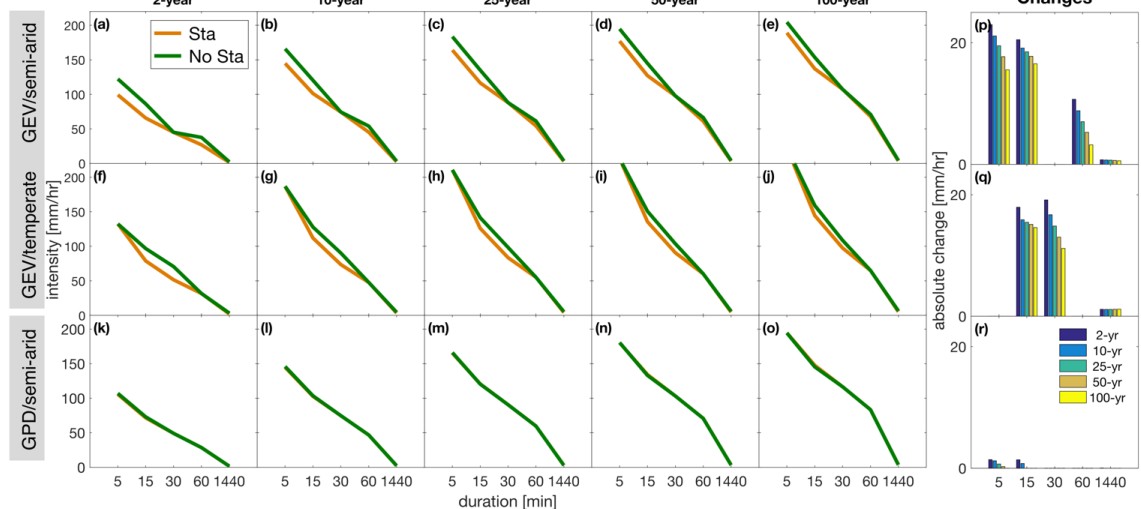

**Figure 6**. Stationary and non-stationary IDF curves for different return periods and durations at the semi-arid (Arizona) and the temperate (Ohio) watersheds. Plots in the right most panels (p. to r.) show changes between estimated non-stationary and stationary intensities as percentage of the stationary estimates (delta= $(X_{non-sta}-X_{sta})/X_{sta} * 100$).





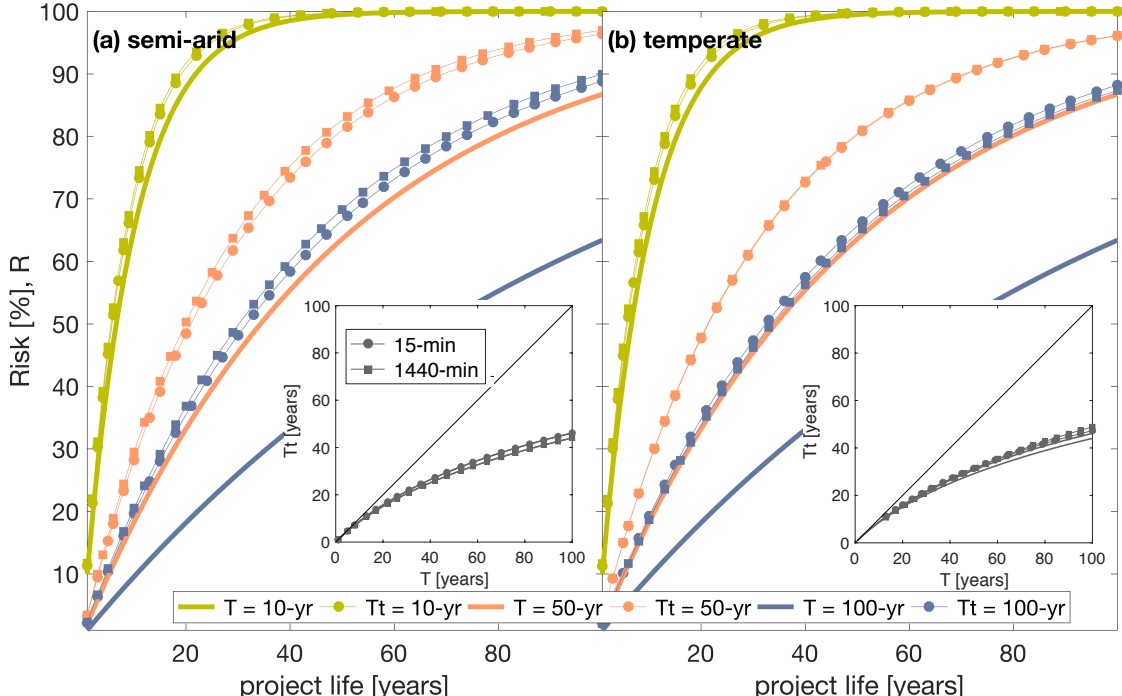

**Figure 7**. Risk of failure as a function of project life (years) assuming stationary return periods (solid lines) T= 10, 50, and 100-year and the corresponding non-stationary Tt return periods (symbols). a. semi-arid site and b. temperate site. In both cases the 15- and 1440-min duration intensities are represented with the GEV model. The subplot inset shows the variation of Tt as a function of T.




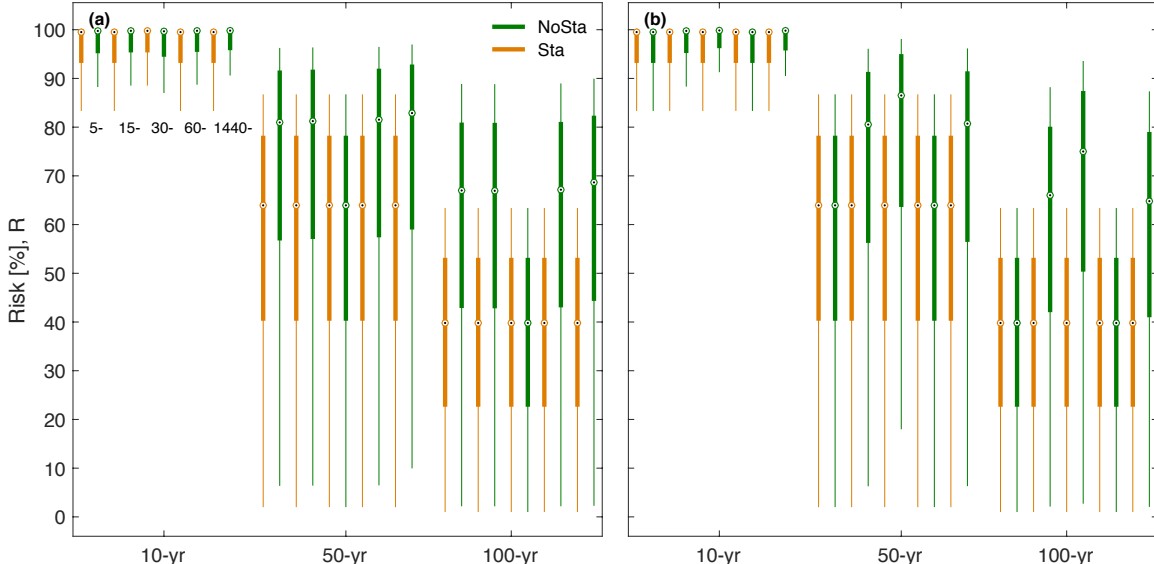

**Figure 8**. Probability of precipitation (risk) for different project life (n: 1 to 100-year) versus return periods in years. Precipitation intensities for durations ranging from 5- to 1440-min are represented with the GEV model. Orange(green) lines show stationary(non-stationary) conditions. a) semi-arid site and b) temperate site. The notch extremes correspond to the 25th and 75th percentiles, respectively, and the extent of the box correspond to the maximum and minimum values.




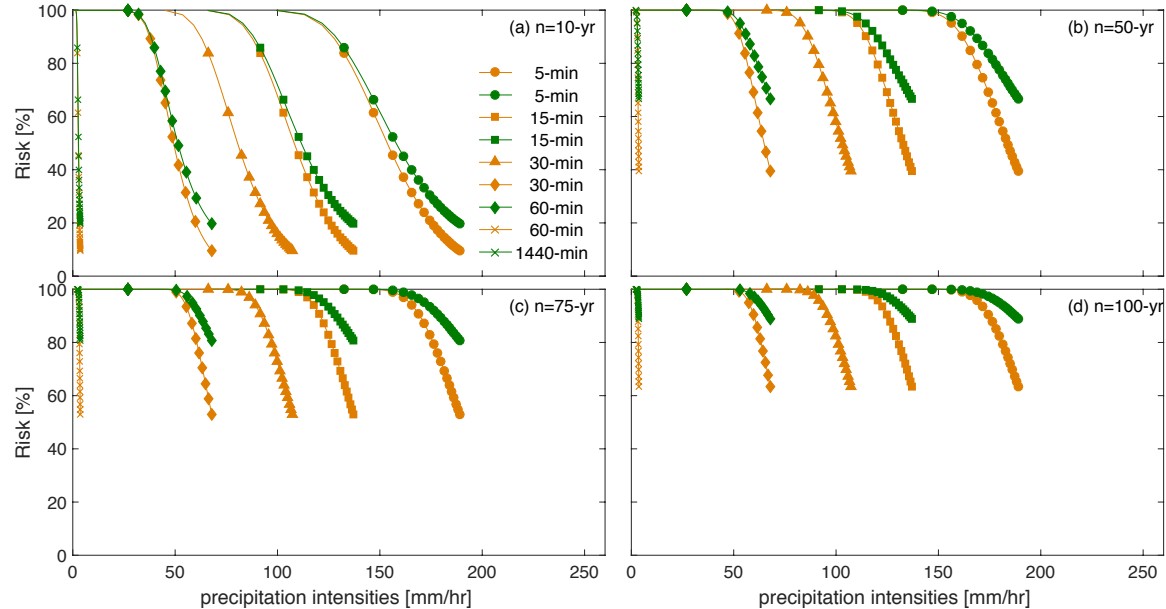

**Figure 9**. Risk of failure as a function of precipitation intensity in the semi-arid site for four project life: a) 10-year, b) 50-year, and c) 75-year and d) 100-year under stationary (orange) and non-stationary (green) conditions. Risk is computed for 1- to 100-yr return periods. Precipitation intensities are represented with the GEV model.

