# Peer review of "Frequency Analysis of Extreme Sub-Daily Precipitation under Stationary and Non-Stationary Conditions across Two Contrasting Hydroclimatic Environments"

_Hydrology and Earth System Sciences, 2017_

## Referee Comment (RC1) · F. Serinaldi (Referee) · 3 Aug 2017

**General comments**

This paper compares Intensity-Duration-Frequency (IDF) curves resulting from stationary and nonstationary frequency analysis. This exercise is not new in the literature, even though the Authors emphasize the quality of the analysed (fine scale) rainfall data set. The statistical methodology devised for such a type of analysis is already available both in frequentist and in Bayesian flavour. In this respect, the paper does not intro-

duce any improvement; on the other hand, I found the application quite confused and questionable. To summarize, in my opinion, this paper is in line with a growing literature adopting some misconceptions and misinterpretations about these topics. In the following I'll provide some remarks, which however are secondary compared with the fact that (non)stationarity is a property of models and cannot be inferred from data only (as done in this paper), thus making these studies (based on trend tests and model selection) no very informative, if not meaningless and misleading. I regret to provide such a negative opinion. I hope that the other reviewers will provide a positive feedback.

**Specific comments**

P2L14-19: the Authors are right when they talk about 'stationarity of the model', but then they confuse stationarity (of the model) with low-frequency fluctuations of ENSO, AMO or whatever climate index as possible sources of nonstationarity. Low-frequency fluctuations resulting in local upward or downward trends (in observed time series) can be due to persistence or other causes. If we do not know the cause of such local trends, they can results from both nonstationary processes and stationary processes (with persistence, regime switch, etc.). The class of stationarity models goes far beyond the i/id case, and comprise (non)i/id processes whose realizations show local trends that can be confused with the effect of nonstationarity. Unfortunately, without identifying the generating mechanism, no definite conclusion can be made about the stationarity or nonstationarity of the underlying process. Concerning the positive increases in observed precipitation linked to a warmer atmosphere, it is worth recalling P1L28-29: the increase of water vapor is a model projection (we can also project vapor decrease, if we want), and the change of the magnitude and frequency of intense precipitation events is a hypothesis (among many others equally reasonable based on the available data).

P2L20: see also Rootzén and Katz (2013) and Serinaldi (2015) for a wider discussion.

P2L22-24: 'A few climate studies have shown that design storm estimates from stationary models are significantly lower than estimates from nonstationary models'. If we assume an increasing trend in the parameters of a distribution (as done in those works and in the present paper), an increase of return level for a given return period or an increase of the risk of failure for a given design value are unavoidable, and trivial, as they are a direct consequence of the model structure. On the other hand, if we incorporate decreasing trends, we should conclude that the estimates from nonstationary models are significantly lower than estimates from stationary models (which is also a trivial conclusion, given the model structure). The point is that we do not the actual direction without identifying the mechanism driving the evolution of the system. Only if we know such a mechanism and we identify a deterministic component, then the use of nonstationary modelling is legitimate; otherwise, it reduces to a simple numerical exercise of parameter fitting on a bunch of data (which is always possible, but uninformative). P2L1-6 and the outcome of the trend tests confirm that we actually have both positive and negative trends: if we fit stationary and nonstationary distributions, we will find that stationary estimates are lower (higher) than nonstationary estimates in the former (latter) case.

P3L21-27: Sincerely, I cannot see any large inter-decadal variability in Fig. 1b; on the contrary, the 11-yr moving average denotes a very regular (flat) behaviour. Please, do not confuse isolated minima and maxima with decadal patterns. Data and resolution are not enough to draw such conclusions, and the diagrams do not support them in any case. The same holds for Fig. 1c: please stop fitting untenable, unjustified, and evidently unsuitable straight lines to hydro-meteorological time series. I hope not to be the only one who see that they are ill-posed and uninformative...maybe not (Poppick et al. 2017). If you do not agree with me, please try this experiment: fit straight lines to data for the periods 1961-1980, 1981-2002, and 1961-2002, then assess the significance of the trend slope, extrapolate to 2015, and finally draw conclusions about the nonstationarity of the process and the design values that we should have adopted. Actually, I think there is no steady increasing pattern, but only a very short time series

whose length is not enough to draw conclusions about the nature of the observed variability, and to support nonstationary models. Concerning sample size requirements and the unreliability of nonstationary design values in future time widows, you can refer to Prosdocimi et al. (2014), Serinaldi and Kilsby (2015), and Luke et al. (2017).

P4L21-26: $T = 1/P$ and $T = 1/rP$ are not definitions but expected values obtained under specific model assumptions (see, Fernández and Salas, 1999; Douglas et al. 2002; Serinaldi 2015; Volpi et al. 2015; Salvadori et al. 2016)

P5L5: for an updated discussion on threshold selection see Langousis et al. (2016) and references therein.

P5L11-14: MK test does not check for linear trends but for stochastic dominance (double check Mann's paper and the rationale Mann-Whitney U-statistic). Moreover, if you check for monotonic trends, why should the trend pattern restricted to the linear one? Why not an S-shaped patter or something else?

P5L20-23: Please, pay attention to cookbook recipes! In their Eq. 10 for equivalent sample size, Yue and Wang (2002) merged the results for lag-1 ACF corresponding to a Markov model (from Matalas and Langbein (1962)) with those for all lags (lag-1 included) from Bayley and Hammersley (1946), which in turn do not rely on Markov assumption. Such hybridization is not required and mixes results relying on different hypotheses. Moreover, ACF estimates are prone to strong bias and required corrections depend (once again) on the assumed model (see, e.g. Koutsoyiannis, 2003; Serinaldi and Kilsby 2016).

P5L25-33: I do not understand which estimation method you use. In these few lines you mention a Bayesian framework without giving any detail, but in the same paragraph, P5L8-10, and in the remainder, you talk about maximum-likelihood estimates, and perform a frequentist analysis based on classical GoF tests, likelihood ratio tests (P6L12-22), etc. Please clarify. My guess is that nothing is Bayesian in this study. If I am wrong, please use a unique approach throughout the paper. Both frequentist

and Bayesian approaches come with a full set of tools for complete inference. Choose the method that you prefer, but avoid exotic hybridizations because there is already enough confusion (and misuse) regarding statistics in hydrology.

P6L1-6: I am sorry, but I think that this is not the correct way to perform this type of analysis. You pretend to compare results from a GEV model where the location parameter (controlling the shift of the distribution) varies linearly with a GPD where the linear behaviour is assigned to the scale parameter related to the variability (spread) of the distribution. This is simply nonsensical: you should choose if the trend is in mean or in variance! Note that you performed trend tests only on the average levels, not on the squared residuals. At most, you should use a GPD with time varying threshold. Moreover, linear trends in the GEV and GPD cannot be justified by the outcome of the trend tests because the distribution introduces a nonlinear transformation between quantiles and location and scale parameters. Again, what means that you multiplied the 40000 parameters by the linear trend? This is not how NS inference works. In the NS framework your distribution is parametrized as $F(x; \mu_0, \mu_1, \sigma, \xi)$; so, your set of MC(MC?) parameters already describes the linear trend and its uncertainty. You do not need to multiply for anything. I only hope that you did not fit the model in a stationary set up, then using the fitted straight lines as correction factors. If so, your approach is not only empirically and theoretical inconsistent, but it also strongly underestimates one of the main sources of uncertainty, i.e., the variability of shape and magnitude of the parameter's trend.

P7L1: Eq. 8 confirms the confusion about the meaning and derivation of the concept of return period: $T_t$ simply does not exist. In both stationary and nonstationary framework, the return period and the corresponding return level are unique value. NS risk of failure can only be written in terms of time-varying probabilities of exceedance $P_t$ because $T$ results from integration (averaging) over a time windows; writing $T_t = 1/P_t$ is meaningless. As mentioned above, $T = 1/P$ is not a definition (axiom) but a relationship derived from calculations under specified model assumptions.

P7L19-31: It seems that the results of trend tests are somewhat mixed with positive and negative trends. So, why do you select two sites with only positive trends, and, in this set, you discard the case showing significant negative trend (temperate, PDS, 60-min)? Leaving aside that the use of trend tests to justify NS models makes little sense, a fair picture should show results for both increasing and decreasing cases. Of course, this would contradict the main conclusion, resulting is a less catchy but more realistic and obvious result: we have both positive and negative trends; they are likely related to random (short-term and long term) fluctuations (or at least the information is not enough to identify a true deterministic evolution), and then NS design values are lower or higher than the stationary counter-part.

P8L9-10: Why are you sure that negative trends are related to natural variability and do not deserve an NS model telling us that the risk is decreasing, whereas positive trends are due to a deterministic mechanism (which is required to justify the NS models) and need NS modelling? Why are negative trends 'children of a lesser god'?

P8L15-24: What means that the theoretical distributions were found statistically significant? Did they pass GoF tests or not? Fig. 3 shows that the fitting is pretty good. Since you have 40000 MC parameter sets, please complement point estimates with confidence intervals (or Bayesian credible intervals).

P8L26-31: The QQ plots in Fig. 4 should correspond to the results in Fig. 3. However, panels in Fig. 4 seem to contain much more data points, and the bad agreement on the upper and lower tails does not match with the good fit shown in Fig. 3. It seems that the QQ plots show results for the stations altogether. I hope to be wrong. Moreover, the temperate site is excluded from QQ plots because it does not show significant trends. Leaving aside that this site shows two significant trends for PDS at 60 and 1440 minutes (see Fig. 2), in P6L10-11 you write that 'This method [QQ plots] was only implemented for the stationary models since the parameters of the non-stationary models change with time'. Thus, you apply QQ plots for stationary cases, but you do not show QQ plots for the stationary case! Is this a three-card Monte
game? :-) By the way, QQ plots can be used in the NS case after suitable rescaling (see, Furrer and Katz, 2008).

P9L1-9: I think that a fairer assessment should include all sites.

P9L15-19: As mentioned above, there is no evidence for any linear trend, and in any case, this does not translate into linear trends in GEV location, and even less in GPD scale parameter. Extrapolating the observed trend is not what NS models do. They extrapolate the law of variation of their parameters with the corresponding uncertainty.

P9L21-23: Please, use transparency to show the overlap of the confidence bands. As mentioned above, obviously NS design values are systematically higher than the stationary values according to the magnitude of the increasing trend introduced in the GEV location parameter. This is a trivial result. Please show also the NS cases with negative trends, where design values decrease compared with the stationary ones.

P9L25-30: 'These results indicate that a specific precipitation intensity will be more likely to occur, i.e., shorter return period, in the future if the observed trends are incorporated in the IDF design.'... For sure, and it will be less likely in the cases of negative trends... and will be unchanged if the trend is null... and water is wet :-) I hope you will understand that the above sentence is pleonastic, actually tautological.

P10L1-12: In text and Fig.6, you refer to absolute differences, but the caption reports the expression of the relative percentage difference. So, what is shown in Fig.6? Which values are discussed in the text? Absolute or relative?

P10L1-12: 'This demonstrates that the stationary framework currently used for structural design systematically underestimate short-duration precipitation extremes which might lead to more frequent infrastructure damage'. I think that this demonstrates that confidence intervals are missing around the point estimates! Add them, and probably the story will change a little bit.

Fig 7: Please add confidence intervals, show cases with negative trends and perform

a fair comparison.

Fig. 8: I cannot see where the variability of the project life appears in these diagrams. Please, clarify.

Fig. 9: Please, add confidence intervals.

**Editing remarks**

Please, check a few typos throughout the text.

Sincerely,

Francesco Serinaldi

**References**

Douglas, E. M., R. M. Vogel, and C. N. Kroll (2002), Impact of streamflow persistence on hydrologic design, J. Hydrol. Eng., 7(3), 220–227 Fernández, B., and J. D. Salas (1999), Return period and risk of hydrologic events. I: mathematical formulation, J. Hydrol. Eng., 4(4), 297–307

Furrer, E.M., and R.W. Katz, 2008: Improving the simulation of extreme precipitation events by stochastic weather generators. Water Resources Research, 44, W12439

Koutsoyiannis D (2003) Climate change, the Hurst phenomenon, and hydrological statistics. Hydrol Sci J 48(1):3–24

Langousis, A., A. Mamalakis, M. Puliga, and R. Deidda (2016), Threshold detection for the generalized Pareto distribution: Review of representative methods and application

to the NOAA NCDC daily rainfall database, Water Resour. Res., 52, 2659–2681

Luke, A., Vrugt, J. A., AghaKouchak, A., Matthew, R., Sanders, B. F., (2017). Predicting nonstationary flood frequencies: Evidence supports an updated stationarity thesis in the United States. Water Resources Research 53.

Poppick, A., Moyer, E. J., Stein, M. L., 2017. Estimating trends in the global mean temperature record. Advances in Statistical Climatology, Meteorology and Oceanography 3 (1), 33-53.

Prosdocimi, I., Kjeldsen, T. R., and Svensson, C. (2014) Non-stationarity in annual and seasonal series of peak flow and precipitation in the UK, Nat. Hazards Earth Syst. Sci., 14, 1125-1144.

Rootzén H, Katz RW (2013) Design life level: quantifying risk in a changing climate. Water Resour Res 49(9):5964–5972

Salvadori, G., F. Durante, C. De Michele, M. Bernardi, and L. Petrella (2016), A multivariate copula-based framework for dealing with hazard scenarios and failure probabilities, Water Resour. Res., 52, 3701–3721

Serinaldi F. Dismissing return periods! (2015). Stochastic Environmental Research and Risk Assessment, 29(4), 1179-1189

Serinaldi F, Kilsby CG. The importance of prewhitening in change point analysis under persistence. Stochastic Environmental Research and Risk Assessment 2016, 30(2), 763-777

Volpi, E., A. Fiori, S. Grimaldi, F. Lombardo, and D. Koutsoyiannis (2015), One hundred years of return period: Strengths and limitations, Water Resour. Res., 51, 8570–8585

---

## Referee Comment (RC2) · Anonymous Referee #2 · 16 Aug 2017

The paper aims to use observed sub-daily summer (June-October) precipitation intensities from two ARS sites to test for evidence of temporal trends and to build IDF curves using Annual Maximum Series (AMS) and Partial Duration Series (PDS) approaches and a Bayesian method that takes into account the non-stationarity of the time series, using this last approach for a failure analysis addressed to infrastructures that are designed with a stationary approach. The paper is interesting and in the line of some recent literature on the topic. There are few major points to be discussed:

1) The proposed Bayesian approach is interesting but some more details should be

provided about the likelihood function and the verification of homoscedastic distribution of residuals. The test is needed if the adopted bayesian likelihood functions is based on such restrictive hypothesis. Otherwise the hypotheses behind the Bayesian approach should be clearly stated (Liuzzo et al. 2017)

2) I understood that Bayesian approach was adopted only for those raingauges for which a local trend was identified (while possible regional crosscorrelation was eliminated by means of RAMK). Bayesian approach is able to generally provide information even if a formal trend cannot be determined providing a sort of "tendency" of the time series to show a trend in the future. This approach can be also replicated in the proposed study with the aim of showing a more general risk analysis

LorenaLiuzzo, VincenzaNotaro, GabrieleFreni (2017) Uncertainty related to climate change in the assessment of the DDF curve parameters. Environmental Modelling & Software Volume 96, October 2017, Pages 1-13

The text is generally well structured and the figures are all informative. I suggest the publication after the comments are addressed.

---

## Author Comment (AC1) · 28 Sep 2017

We thank the Reviewers for their valuable and useful comments on this manuscript. Their suggestions will further improve our manuscript and we are certain that we can address the comments in the revised manuscript. Many of the Reviewers' comments highlight the complexity of the problem this paper seeks to address. We hope that this manuscript will add clarity on how storm design and risk might change in the changing world we live in. Please see below our response to each of the reviewers' comment.

[Figure]
General comments This paper compares Intensity-Duration-Frequency (IDF) curves resulting from stationary and nonstationary frequency analysis. This exercise is not new in the literature, even though the Authors emphasize the quality of the analysed (fine scale) rainfall data set. The statistical methodology devised for such a type of analysis is already available both in frequentist and in Bayesian flavour. In this respect, the paper does not introduce any improvement; on the other hand, I found the application quite confused and questionable. To summarize, in my opinion, this paper is in line with a growing literature adopting some misconceptions and misinterpretations about these topics. In the following I'll provide some remarks, which however are secondary compared with the fact that (non)stationarity is a property of models and cannot be inferred from data only (as done in this paper), thus making these studies (based on trend tests and model selection) no very informative, if not meaningless and misleading. I regret to provide such a negative opinion. I hope that the other reviewers will provide a positive feedback.

We thank the Dr. Serinaldi for the valuable and useful comments on this manuscript, they have certainly helped improve our understanding of the statistical concepts we are using in our study. We believe that their suggestions will further improve our manuscript and we can address these comments in the revised manuscript. We understand the Reviewer's concerns and we are aware of the available scientific literature that reflects on the misuse of statistics done by hydrologists (Clarke, 2010;WMO, 2013;Koutsoyiannis and Montanari, 2015;Lins and Cohn, 2011). However, despite the limitations of our approach we think that there is considerable scientific evidence, albeit uncertain, that

precipitation intensities have increased with warmer atmospheric temperatures (Fischer and Knutti, 2016;Lenderink and van Meijgaard, 2008;Mishra et al., 2012;Muschinski and Katz, 2013;Wasko and Sharma, 2015;Wasko et al., 2016;Westra et al., 2014) and these increases need to be taken into account in the design process. The authors do not think that a measure of humility (Lins and Cohn, 2011) is enough to design structures and to implement water management policies that allow a sustainable improvement of people's quality of life. In many localities, sub-daily duration design storms are specified for design of storm water management structures. The dataset employed in our analysis is not optimal for length of record (few are if the criteria of the reviewer is used), but nonetheless, the analysis of this data provides valuable insights into sub-daily intensities and the differences in their trends as compared to daily analysis for which the vast majority of intensity trend analysis has been conducted.

Specific comments

P2L14-19: the Authors are right when they talk about 'stationarity of the model', but then they confuse stationarity (of the model) with low-frequency fluctuations of ENSO, AMO or whatever climate index as possible sources of nonstationarity. Low-frequency fluctuations resulting in local upward or downward trends (in observed time series) can be due to persistence or other causes. If we do not know the cause of such local trends, they can results from both nonstationary processes and stationary processes (with persistence, regime switch, etc.). The class of stationarity models goes far beyond the i/id case, and comprise (non)i/id processes whose realizations show local trends that can be confused with the effect of nonstationarity. Unfortunately, without identifying the generating mechanism, no definite conclusion can be made about the stationarity or nonstationarity of the underlying process.

The Reviewer raises an excellent point. We consider that natural systems are inherently non-stationary with hydro-climatological time series having falling and rising local trends known as long-term persistence or Hurst's phenomena (Koutsoyiannis, 2005). In the text (P2L15), we refer to them as low frequency components of climate variability

such as ENSO, AMO, PDO. The presence of long-term persistence can result in the detection of statistically significant trends even when no trend is present (Cohn and Lins, 2005;Villarini et al., 2009). We are aware than from an statistical perspective even the longest available record of a climate variable can exhibit long-term persistence when an spectral analysis, which is also a model with limitations and assumptions, is performed on them (Lins and Cohn, 2011). However, the difficulty identifying the forcing mechanism generating such trends can be challenging due to the length of the available observational records time series, and it might be easier, as Villarini et al. (2009) suggested, to proclaim the demise of stationarity than to prove it with the available data.

After reading Dr. Serinaldi's comments, we realized that our wording is incorrect and it needs to be improved in the revised version of the manuscript. Specifically, the text should read (P2L14) stationary of the time series not stationarity of the model. We will revise this section in the revised manuscript and provide additional information about the limitations of our approach.

Concerning the positive increases in observed precipitation linked to a warmer atmosphere, it is worth recalling P1L28-29: the increase of water vapor is a model projection (we can also project vapor decrease, if we want), and the change of the magnitude and frequency of intense precipitation events is a hypothesis (among many others equally reasonable based on the available data).

We agree that climate models are imperfect representations of the complex climate system. Their uncertainties are constantly being evaluated by the scientific community and improved model versions are being developed as computational power increases. However, we believe that despite their limitations, climate models constitute the best available tool to reproduce past and future climates. Stochastic models are excellent options but they are black box approaches that rely only in the memory of the data available. Climate models on the other hand, use physically-based equations, when feasible, to represent natural processes in the ocean, atmosphere, and land surface.

Additionally to climate model simulations, precipitation and temperature observations during the last half of the 20th century show, in the absence of changes in circulation patterns, that precipitation intensities have increased at ratios that exceeded the expected from the Clausius-Clapeyron relation which describes the capacity of the atmosphere to hold moisture (Fischer and Knutti, 2016;Lenderink and van Meijgaard, 2008;Mishra et al., 2012;Wasko et al., 2016). Within virtually all of the scientifically informed climate community there is clear consensus that atmospheric temperatures are increasing, as show within the observational record with increasing greenhouse gases. It is a matter of simple physics. As projected trends in greenhouse gas releases are increasing for some time, even under the most optimistic reduction scenarios, it is safe to assume temperatures will also increase. The physics clearly indicates that a warmer atmosphere can contain greater water vapor so the projection of increasing atmospheric water vapor is hardly a contentious assertion.

P2L20: see also Rootzén and Katz (2013) and Serinaldi (2015) for a wider discussion. Thank you for referencing these papers. We will incorporate them in the revised version of the manuscript.

P2L22-24: 'A few climate studies have shown that design storm estimates from stationary models are significantly lower than estimates from nonstationary models'. If we assume an increasing trend in the parameters of a distribution (as done in those works and in the present paper), an increase of return level for a given return period or an increase of the risk of failure for a given design value are unavoidable, and trivial, as they are a direct consequence of the model structure. On the other hand, if we incorporate decreasing trends, we should conclude that the estimates from nonstationary models are significantly lower than estimates from stationary models (which is also a trivial conclusion, given the model structure). The point is that we do not the actual direction without identifying the mechanism driving the evolution of the system. Only if we know such a mechanism and we identify a deterministic component, then the use of nonstationary modelling is legitimate; otherwise, it reduces to a simple numerical exercise of

parameter fitting on a bunch of data (which is always possible, but uninformative).

This is a very interesting point. The mechanism driving the trends is not well understood and should be further investigated. In the semi-arid site winter, spring and fall precipitation are enhanced under El Nino-Southern Oscillation (ENSO) conditions and summer precipitation is reduced (Webb and Betancourt, 1992). Annual droughts are more frequent when positive Atlantic Multidecadal Oscillation (AMO) and negative Pacific Decadal Oscillation (PDO) occur (McCabe et al., 2004). In the temperate site, wetter conditions during winter have been dominated by El Niño condition with the high-Pacific-North American pattern (PNA) andÂăPacific Decadal Oscillation (PDO) indices (Ning and Bradley, 2014).

P2L1-6 and the outcome of the trend tests confirm that we actually have both positive and negative trends: if we fit stationary and nonstationary distributions, we will find that stationary estimates are lower (higher) than nonstationary estimates in the former (latter) case.

This pattern of positive and negative trends in precipitation has been reported in the literature. We agree that the lower/higher estimates between stationary/nonstationary models is expected, however engineering design needs an estimate, albeit uncertain, of the magnitude of those ups and downs. Saying that the estimates are lower or higher than the available design storms are it is useless for informing structural design.

P3L21-27: Sincerely, I cannot see any large inter-decadal variability in Fig. 1b; on the contrary, the 11-yr moving average denotes a very regular (flat) behaviour. Please, do not confuse isolated minima and maxima with decadal patterns. Data and resolution are not enough to draw such conclusions, and the diagrams do not support them in any case. The same holds for Fig. 1c: please stop fitting untenable, unjustified, and evidently unsuitable straight lines to hydro-meteorological time series. I hope not to be the only one who see that they are ill-posed and uninformative...maybe not (Poppick et al. 2017). If you do not agree with me, please try this experiment: fit straight
lines to data for the periods 1961-1980, 1981-2002, and 1961-2002, then assess the significance of the trend slope, extrapolate to 2015, and finally draw conclusions about the nonstationarity of the process and the design values that we should have adopted. Actually, I think there is no steady increasing pattern, but only a very short time series whose length is not enough to draw conclusions about the nature of the observed variability, and to support nonstationary models. Concerning sample size requirements and the unreliability of nonstationary design values in future time widows, you can refer to Prosdocimi et al. (2014), Serinaldi and Kilsby (2015), and Luke et al. (2017).

Thank you for your comments. The Reviewer is correct, based on the figure alone it is not possible to infer the presence of inter-decadal variability. We will rewrite the description of Fig 1 (b) and (c) to incorporate this comment.

Regarding the fitting of a straight line to the figures, the Reviewer is correct to point out that the magnitude and direction of the trend is dependent on the beginning and ending years and has been clearly demonstrated with different periods of the Southwest USA observations (Goodrich et al., 2008). We agree with the Reviewer that fitting a straight line to a different period might change the results. In the revised manuscript, we will add a Block Bootstrapping Mann-Kendall (BBS-MK) trend analysis to examine if the rainfall intensities for different time series lengths show a nonstationary behavior. In the BBS-MK approach (Kundzewicz and Robson, 2004), the original time series are randomly resampled in blocks for a set period of time which allows the original short-term memory of the system to be preserved. The method is flexible and robust and avoids any modification of the original dependence structure of the data which constitutes one of the strengths of the method.

P4L21-26: T = 1/P and T = 1/rP are not definitions but expected values obtained under specific model assumptions (see, Fernández and Salas, 1999; Douglas et al. 2002; Serinaldi 2015; Volpi et al. 2015; Salvadori et al. 2016) Thank you pointing this out. We will replace the equations in the manuscript with Expected values T=E(x)=.... We agree with the Reviewer that the traditional (stationary) approach for defining T is based

on the assumption of serially independent annual maximum events that follow a geometric distribution with an exceedance probability p. Under stationary conditions, the mean return period is always 1/p; however under non-stationary conditions p is no longer constant from year to year (Read and Vogel, 2015;Vogel et al., 2015). The approach we are using (Non-stationary Extreme Value Analysis (NEVA), (Cheng and AghaKouchak, 2014;Cheng et al., 2014)), computes time varying exceedance probabilities as outlined in (Salas and Obeysekera, 2014).

P5L5: for an updated discussion on threshold selection see Langousis et al. (2016) and references therein. We will improve the discussion in the revised version of the manuscript.

P5L11-14: MK test does not check for linear trends but for stochastic dominance (double check Mann's paper and the rationale Mann-Whitney U-statistic). Moreover, if you check for monotonic trends, why should the trend pattern restricted to the linear one? Why not an S-shaped patter or something else?

The Reviewer is right, the Mann test is a test of randomness against trend. Why linear? That is a very good point, a linear trend is frequently chosen for hydrologic and climatic analysis hence we chose it to be able to compare our results with similar analysis available in the literature (Groisman et al., 2005;Kunkel et al., 1999;Kunkel et al., 2012;Pryor et al., 2009) among others).

P5L20-23: Please, pay attention to cookbook recipes! In their Eq. 10 for equivalent sample size, Yue and Wang (2002) merged the results for lag-1 ACF corresponding to a Markov model (from Matalas and Langbein (1962)) with those for all lags (lag-1 included) from Bayley and Hammersley (1946), which in turn do not rely on Markov assumption. Such hybridization is not required and mixes results relying on different hypotheses. Moreover, ACF estimates are prone to strong bias and required corrections depend (once again) on the assumed model (see, e.g. Koutsoyiannis, 2003; Serinaldi and Kilsby 2016).

We thank the Reviewer for pointing this out. We are not sure what the Reviewer is referring to with cookbook recipes. We have implemented Eq. 10 from Yue and Wang (2002) as indicated by the authors.

P5L25-33: I do not understand which estimation method you use. In these few lines you mention a Bayesian framework without giving any detail, but in the same paragraph, P5L8-10, and in the remainder, you talk about maximum-likelihood estimates, and perform a frequentist analysis based on classical GoF tests, likelihood ratio tests (P6L12-22), etc. Please clarify. My guess is that nothing is Bayesian in this study. If I am wrong, please use a unique approach throughout the paper. Both frequentist and Bayesian approaches come with a full set of tools for complete inference. Choose the method that you prefer, but avoid exotic hybridizations because there is already enough confusion (and misuse) regarding statistics in hydrology.

We appreciate the Reviewer's comment. We are using a Bayesian-based Markov chain Monte Carlo approach to obtain the posterior distribution of parameters from an arbitrary distribution of the parameter space. First the prior parameter values are specified with an assumed probability distribution (Cheng et al., 2014), then assuming that the observations are independent the Bayes theorem is used to compute the posterior parameter distribution.

The Reviewer is correct about using the ML method to fit the GEV and GPD models. We realized that the statement in P5L5-10 is incorrect. We used the ML method to estimate the GPD threshold parameter under stationary assumptions. The goodness of both probabilistic models was measured with the log-likelihood ratio method was used to compare the fit of the stationary and nonstationary models (Coles, 2001).

We will include additional information about the methods used and remove confusing information in the revised manuscript.

P6L1-6: I am sorry, but I think that this is not the correct way to perform this type of analysis. You pretend to compare results from a GEV model where the location

parameter (controlling the shift of the distribution) varies linearly with a GPD where the linear behaviour is assigned to the scale parameter related to the variability (spread) of the distribution. This is simply nonsensical: you should choose if the trend is in mean or in variance! Note that you performed trend tests only on the average levels, not on the squared residuals. At most, you should use a GPD with time varying threshold. Moreover, linear trends in the GEV and GPD cannot be justified by the outcome of the trend tests because the distribution introduces a nonlinear transformation between quantiles and location and scale parameters. Again, what means that you multiplied the 40000 parameters by the linear trend? This is not how NS inference works. In the NS framework your distribution is parametrized as $F(x; \mu0, \mu1, \sigma, \xi)$; so, your set of MC(MC?) parameters already describes the linear trend and its uncertainty. You do not need to multiply for anything. I only hope that you did not fit the model in a stationary set up, then using the fitted straight lines as correction factors. If so, your approach is not only empirically and theoretical inconsistent, but it also strongly underestimates one of the main sources of uncertainty, i.e., the variability of shape and magnitude of the parameter's trend.

We included the GPD scale parameter () based on recommendations by (Cheng, personal communication and (Cheng et al., 2014)) who indicated that long-term time series are required to model the temporal changes in this parameter. We do not think that our approach is nonsensical, we are clearly stating what parameters were included in the analysis and that comparing side-by-side GEV to GPD is not one of the goals of the study. That said, we realized that this is not clear in the manuscript in its current form and we will incorporate the Reviewer's comments in a revised version of it.

The Reviewer suggests "At most, you should use a GPD with time varying threshold", we chose to keep the threshold constant which will allows us to evaluate if precipitation intensities have increased/decreased in time during the study period. By allowing the threshold parameter to move in time, it is not possible to have a benchmark for that selection.

We realized that our wording was unclear and misleading. We did not fit the GEV/GPD model in a stationary setup and added the trend line. For each Monte Carlo realization (40000), the time-variant location parameter (in the case of the GEV) linearly varies in time ((t) = 1t + 0 values). The model parameters are used to estimate nonstationary precipitation intensities for different exceedance probabilities (AghaKouchak et al., 2013;Cheng and AghaKouchak, 2014;Cheng et al., 2014).

We will modify the text in the revised manuscript to reflect this comment.

P7L1: Eq. 8 confirms the confusion about the meaning and derivation of the concept of return period: Tt simply does not exist. In both stationary and nonstationary framework, the return period and the corresponding return level are unique value. NS risk of failure can only be written in terms of time-varying probabilities of exceedance Pt because T results from integration (averaging) over a time windows; writing Tt = 1/Pt is meaningless. As mentioned above, T = 1/P is not a definition (axiom) but a relationship derived from calculations under specified model assumptions.

We are not sure what the Reviewer means by "Tt simply does not exist". In our study the nonstationary return periods are computed with the expected waiting time theory. Details of the approach are found in (Cheng and AghaKouchak, 2014;Salas and Obeysekera, 2014). As we mentioned in a previous comment, we are aware that T is dependent on model assumptions (usually a geometric distribution under stationary conditions) that are not valid for non-stationary conditions (Read and Vogel, 2015;Vogel et al., 2015). The approach we are using (Non-stationary Extreme Value Analysis (NEVA), (Cheng and AghaKouchak, 2014;Cheng et al., 2014)), computes time varying exceedance probabilities as outlined in (Salas and Obeysekera, 2014).

P7L19-31: It seems that the results of trend tests are somewhat mixed with positive and negative trends. So, why do you select two sites with only positive trends, and, in this set, you discard the case showing significant negative trend (temperate, PDS, 60-min)? Leaving aside that the use of trend tests to justify NS models makes little

sense, a fair picture should show results for both increasing and decreasing cases. Of course, this would contradict the main conclusion, resulting is a less catchy but more realistic and obvious result: we have both positive and negative trends; they are likely related to random (short-term and long term) fluctuations (or at least the information is not enough to identify a true deterministic evolution), and then NS design values are lower or higher than the stationary counter-part.

This is a very good point. We selected only positive trends because we consider that from a structural design perspective, the consequences of underestimating the design storms are more damaging to society than overdesigning them. We agree that including negative trends will most likely result in underestimations of precipitation intensities. We think that incorporating the negative trends in our analysis will be an interesting exercise and will improve the manuscript. To show the differences, we will implement the approach using negative trends in the revised manuscript.

P8L9-10: Why are you sure that negative trends are related to natural variability and do not deserve an NS model telling us that the risk is decreasing, whereas positive trends are due to a deterministic mechanism (which is required to justify the NS models) and need NS modelling? Why are negative trends 'children of a lesser god'?

This comment is related to the previous one. Negative trends are not "children of a lesser god", we considered that their impact on engineering design was not as crucial as in the case of increasing trends.

P8L15-24: What means that the theoretical distributions were found statistically significant? Did they pass GoF tests or not? Fig. 3 shows that the fitting is pretty good. Since you have 40000 MC parameter sets, please complement point estimates with confidence intervals (or Bayesian credible intervals).

The GoF for the stationary models is very good and the fitting passed the test. This is shown in Figure 3. We are not plotting the fitting for the NS models in this figure.

P8L26-31: The QQ plots in Fig. 4 should correspond to the results in Fig. 3. However, panels in Fig. 4 seem to contain much more data points, and the bad agreement on the upper and lower tails does not match with the good fit shown in Fig. 3. It seems that the QQ plots show results for the stations altogether. I hope to be wrong. Moreover, the temperate site is excluded from QQ plots because it does not show significant trends. Leaving aside that this site shows two significant trends for PDS at 60 and 1440 minutes (see Fig. 2), in P6L10-11 you write that 'This method [QQ plots] was only implemented for the stationary models since the parameters of the non-stationary models change with time'. Thus, you apply QQ plots for stationary cases, but you do not show QQ plots for the stationary case! Is this a three-card Montegame? :-) By the way, QQ plots can be used in the NS case after suitable rescaling (see, Furrer and Katz, 2008).

We will verify that the data used to plot Figures 3 and 4 are consistent. We didn't include the temperature site in the Q-Q plots in Figure 4 because we didn't use the site for further analysis in the paper. We will include the data in the revised manuscript. We are not sure we understand the Reviewer's comment "Thus, you apply QQ plots for stationary cases, but you do not show QQ plots for the stationary case! Is this a three-card Montegame? :-)". Perhaps he meant nonstationary case. We didn't show the Q-Q plots for NS models, we will include them in the revised version of the manuscript.

P9L1-9: I think that a fairer assessment should include all sites. We will include all sites in the analysis in the revised manuscript.

P9L15-19: As mentioned above, there is no evidence for any linear trend, and in any case, this does not translate into linear trends in GEV location, and even less in GPD scale parameter. Extrapolating the observed trend is not what NS models do. They extrapolate the law of variation of their parameters with the corresponding uncertainty.

We extrapolate the observed linear trend in our models since we assume that the rate of change in precipitation intensities will stay constant in the future. Based on

this assumption, the GEV location parameter and the GPD scale parameter will vary accordingly. Perhaps this is an approach with its limitations, however is widely used in the climate and hydrologic sciences (Condon et al., 2015;Liuzzo et al., 2017;Luke et al., 2017;Sarhadi and Soulis, 2017;Wi et al., 2016). The goal of this paper is to use available statistical tools to evaluate changes in design storms not to develop new ones.

P9L21-23: Please, use transparency to show the overlap of the confidence bands. As mentioned above, obviously NS design values are systematically higher than the stationary values according to the magnitude of the increasing trend introduced in the GEV location parameter. This is a trivial result. Please show also the NS cases with negative trends, where design values decrease compared with the stationary ones.

We will use transparent confidence bands and negative trends in the revised manuscript.

P9L25-30: 'These results indicate that a specific precipitation intensity will be more likely to occur, i.e., shorter return period, in the future if the observed trends are incorporated in the IDF design.'... For sure, and it will be less likely in the cases of negative trends. . . and will be unchanged if the trend is null... and water is wet :-) I hope you will understand that the above sentence is pleonastic, actually tautological.

We will include the analysis with negative trends in the revised version to be able to quantify the magnitude of the changes. We do not consider our work superfluous or we would not be doing it. We will improve the writing in the revised manuscript to avoid needless repetition of ideas.

P10L1-12: In text and Fig.6, you refer to absolute differences, but the caption reports the expression of the relative percentage difference. So, what is shown in Fig.6? Which values are discussed in the text? Absolute or relative?

We thank the Reviewer for pointing this out. The label is incorrect, it is absolute

changes. We will correct this mistake in the revised manuscript.

P10L1-12: 'This demonstrates that the stationary framework currently used for structural design systematically underestimate short-duration precipitation extremes which might lead to more frequent infrastructure damage'. I think that this demonstrates that confidence intervals are missing around the point estimates! Add them, and probably the story will change a little bit. Fig 7: Please add confidence intervals, show cases with negative trends and perform a fair comparison. Fig. 8: I cannot see where the variability of the project life appears in these diagrams. Please, clarify. Fig. 9: Please, add confidence intervals.

We will include all these comments in the new manuscript.

Editing remarks Please, check a few typos throughout the text.

Sincerely, Francesco Serinaldi

References Douglas, E. M., R. M. Vogel, and C. N. Kroll (2002), Impact of streamflow persistence on hydrologic design, J. Hydrol. Eng., 7(3), 220–227

Fernández, B., and J. D. Salas (1999), Return period and risk of hydrologic events. I: mathematical formulation, J. Hydrol. Eng., 4(4), 297–307

Furrer, E.M., and R.W. Katz, 2008: Improving the simulation of extreme precipitation events by stochastic weather generators. Water Resources Research, 44, W12439

Koutsoyiannis D (2003) Climate change, the Hurst phenomenon, and hydrological statistics. Hydrol Sci J 48(1):3–24

Langousis, A., A. Mamalakis, M. Puliga, and R. Deidda (2016), Threshold detection for the generalized Pareto distribution: Review of representative methods and application to the NOAA NCDC daily rainfall database, Water Resour. Res., 52, 2659–2681

Luke, A., Vrugt, J. A., AghaKouchak, A., Matthew, R., Sanders, B. F., (2017). Predicting nonstationary flood frequencies: Evidence supports an updated stationarity thesis in

the United States. Water Resources Research 53.

Poppick, A., Moyer, E. J., Stein, M. L., 2017. Estimating trends in the global mean temperature record. Advances in Statistical Climatology, Meteorology and Oceanography 3 (1), 33-53.

Prosdocimi, I., Kjeldsen, T. R., and Svensson, C. (2014) Non-stationarity in annual and seasonal series of peak flow and precipitation in the UK, Nat. Hazards Earth Syst. Sci., 14, 1125-1144.

Rootzén H, Katz RW (2013) Design life level: quantifying risk in a changing climate. Water Resour Res 49(9):5964–5972

Salvadori, G., F. Durante, C. De Michele, M. Bernardi, and L. Petrella (2016), A multivariate copula-based framework for dealing with hazard scenarios and failure probabilities, Water Resour. Res., 52, 3701–3721

Serinaldi F. Dismissing return periods! (2015). Stochastic Environmental Research and Risk Assessment, 29(4), 1179-1189

Serinaldi F, Kilsby CG. The importance of prewhitening in change point analysis under persistence. Stochastic Environmental Research and Risk Assessment 2016, 30(2), 763-777

Volpi, E., A. Fiori, S. Grimaldi, F. Lombardo, and D. Koutsoyiannis (2015), One hundred years of return period: Strengths and limitations, Water Resour. Res., 51, 8570–8585 Interactive comment on Hydrol. Earth Syst. Sci. Discuss., https://doi.org/10.5194/hess-2017- 247, 2017.

References: AghaKouchak, A., Easterling, D., Hsu, K., Schubert, S., and Sorooshian, S.: Extremes in a Changing Climate. Detection, Analysis and Uncertainty, 1 ed., edited by: AghaKouchak, A., Easterling, D., Hsu, K., Schubert, S., and Sorooshian, S., Springer Netherlands , 2013.

Cheng, L., and AghaKouchak, A.: Nonstationary Precipitation Intensity-Duration-Frequency Curves for Infrastructure Design in a Changing Climate, 4, 7093, 10.1038/srep07093 https://www.nature.com/articles/srep07093 - supplementary-information, 2014.

Cheng, L., AghaKouchak, A., Gilleland, E., and Katz, R. W.: Non-stationary extreme value analysis in a changing climate, Climatic Change, 127, 353-369, 10.1007/s10584-014-1254-5, 2014.

Clarke, R. T.: On the (mis)use of statistical methods in hydro-climatological research, Hydrological Sciences Journal, 55, 139-144, 10.1080/02626661003616819, 2010.

Cohn, T. A., and Lins, H. F.: Nature's style: Naturally trendy, Geophysical Research Letters, 32, 10.1029/2005GL024476, 2005.

Coles, S.: An Introduction to Statistical Modeling of Extreme Values, edited by: London, S., London, 2001.

Condon, L. E., Gangopadhyay, S., and Pruitt, T.: Climate change and non-stationary flood risk for the upper Truckee River basin, Hydrol. Earth Syst. Sci., 19, 159-175, 10.5194/hess-19-159-2015, 2015.

Fischer, E. M., and Knutti, R.: Observed heavy precipitation increase confirms theory and early models, Nature Clim. Change, 6, 986-991, 10.1038/nclimate3110 http://www.nature.com/nclimate/journal/v6/n11/abs/nclimate3110.html - supplementary-information, 2016.

Goodrich, D. C., Unkrich, C. L., Keefer, T. O., Nichols, M. H., Stone, J. J., Levick, L. R., and Scott, R. L.: Event to multidecadal persistence in rainfall and runoff in southeast Arizona, Water Resources Research, 44, n/a-n/a, 10.1029/2007WR006222, 2008.

Groisman, P. Y., Knight, R. W., Easterling, D. R., Karl, T. R., Hegerl, G. C., and Razuvaev, V. N.: Trends in Intense Precipitation in the Climate Record, Journal of Climate, 18, 1326-1350, 10.1175/JCLI3339.1, 2005.

Koutsoyiannis, D.: Hydrologic Persistence and The Hurst Phenomenon, in: Water Encyclopedia, 210–221, 2005.

Koutsoyiannis, D., and Montanari, A.: Negligent killing of scientific concepts: the stationarity case, Hydrological Sciences Journal, 60, 1174-1183, 10.1080/02626667.2014.959959, 2015.

Kundzewicz, Z. W., and Robson, A. J.: Change detection in hydrological records—a review of the methodology / Revue méthodologique de la détection de changements dans les chroniques hydrologiques, Hydrological Sciences Journal, 49, 7-19, 10.1623/hysj.49.1.7.53993, 2004.

Kunkel, K. E., Andsager, K., and Easterling, D. R.: Long-Term Trends in Extreme Precipitation Events over the Conterminous United States and Canada, Journal of Climate, 12, 2515-2527, 10.1175/1520-0442(1999)012<2515:LTTIEP>2.0.CO;2, 1999.

Kunkel, K. E., Easterling, D. R., Kristovich, D. A. R., Gleason, B., Stoecker, L., and Smith, R.: Meteorological Causes of the Secular Variations in Observed Extreme Precipitation Events for the Conterminous United States, Journal of Hydrometeorology, 13, 1131-1141, 10.1175/jhm-d-11-0108.1, 2012.

Lenderink, G., and van Meijgaard, E.: Increase in hourly precipitation extremes beyond expectations from temperature changes, Nature Geosci, 1, 511-514, http://www.nature.com/ngeo/journal/v1/n8/suppinfo/ngeo262_S1.html, 2008.

Lins, H. F., and Cohn, T. A.: Stationarity: Wanted Dead or Alive?1, JAWRA Journal of the American Water Resources Association, 47, 475-480, 10.1111/j.1752-1688.2011.00542.x, 2011. Liuzzo, L., Notaro, V., and Freni, G.: Uncertainty related to climate change in the assessment of the DDF curve parameters, Environmental Modelling & Software, 96, 1-13, https://doi.org/10.1016/j.envsoft.2017.06.044, 2017.

Luke, A., Vrugt, J. A., AghaKouchak, A., Matthew, R., and Sanders, B. F.: Predicting nonstationary flood frequencies: Evidence supports an updated stationarity thesis in the United States, Water Resources Research, 53, 5469-5494, 10.1002/2016WR019676, 2017.

McCabe, G. J., Palecki, M. A., and Betancourt, J. L.: Pacific and Atlantic Ocean influences on multidecadal drought frequency in the United States, Proceedings of the National Academy of Sciences, 101, 4136-4141, 10.1073/pnas.0306738101, 2004.

Mishra, V., Wallace, J. M., and Lettenmaier, D. P.: Relationship between hourly extreme precipitation and local air temperature in the United States, Geophysical Research Letters, 39, n/a-n/a, 10.1029/2012GL052790, 2012.

Muschinski, T., and Katz, J. I.: Trends in hourly rainfall statistics in the United States under a warming climate, Nature Clim. Change, 3, 577-580, 10.1038/nclimate1828, 2013.

Ning, L., and Bradley, R. S.: Winter precipitation variability and corresponding teleconnections over the northeastern United States, Journal of Geophysical Research: Atmospheres, 119, 7931-7945, 10.1002/2014JD021591, 2014.

Pryor, S. C., Howe, J. A., and Kunkel, K. E.: How spatially coherent and statistically robust are temporal changes in extreme precipitation in the contiguous USA?, International Journal of Climatology, 29, 31-45, 10.1002/joc.1696, 2009.

Read, L. K., and Vogel, R. M.: Reliability, return periods, and risk under nonstationarity, Water Resources Research, 51, 6381-6398, 10.1002/2015WR017089, 2015.

Salas, J. D., and Obeysekera, J.: Revisiting the Concepts of Return Period and Risk for Nonstationary Hydrologic Extreme Events, Journal of Hydrologic Engineering, 19, 554-568, doi:10.1061/(ASCE)HE.1943-5584.0000820, 2014.

Sarhadi, A., and Soulis, E. D.: Time-varying extreme rainfall intensity-duration-frequency curves in a changing climate, Geophysical Research Letters, 44, 2454-2463, 10.1002/2016GL072201, 2017.

Villarini, G., Serinaldi, F., Smith, J. A., and Krajewski, W. F.: On the stationarity of annual flood peaks in the continental United States during the 20th century, Water Resources Research, 45, n/a-n/a, 10.1029/2008WR007645, 2009.

Vogel, R. M., Lall, U., Cai, X., Rajagopalan, B., Weiskel, P. K., Hooper, R. P., and Matalas, N. C.: Hydrology: The interdisciplinary science of water, Water Resources Research, 51, 4409-4430, 10.1002/2015WR017049, 2015.

Wasko, C., and Sharma, A.: Steeper temporal distribution of rain intensity at higher temperatures within Australian storms, Nature Geosci, 8, 527-529, 10.1038/ngeo2456 http://www.nature.com/ngeo/journal/v8/n7/abs/ngeo2456.html - supplementary-information, 2015.

Wasko, C., Sharma, A., and Westra, S.: Reduced spatial extent of extreme storms at higher temperatures, Geophysical Research Letters, 43, 4026-4032, 10.1002/2016GL068509, 2016.

Webb, R. H., and Betancourt, J. L.: Climatic variability and flood frequency of the Santa Cruz River, Pima County, Arizona, 1992.

Westra, S., Fowler, H. J., Evans, J. P., Alexander, L. V., Berg, P., Johnson, F., Kendon, E. J., Lenderink, G., and Roberts, N. M.: Future changes to the intensity and frequency of short-duration extreme rainfall, Reviews of Geophysics, 52, 522-555, 10.1002/2014RG000464, 2014.

Wi, S., Valdés, J. B., Steinschneider, S., and Kim, T.-W.: Non-stationary frequency analysis of extreme precipitation in South Korea using peaks-over-threshold and annual maxima, Stochastic Environmental Research and Risk Assessment, 30, 583-606, 10.1007/s00477-015-1180-8, 2016.

WMO: Commission for Hydrology, CHy Statement on the Terms Stationarity and Non-stationarity, World Meteorological Organization, 2013.

Yue, S., and Wang, C. Y.: Regional streamflow trend detection with consideration

of both temporal and spatial correlation, International Journal of Climatology, 22, 933-946, 10.1002/joc.781, 2002.

Please also note the supplement to this comment: https://www.hydrol-earth-syst-sci-discuss.net/hess-2017-247/hess-2017-247-AC1-supplement.pdf

---

## Author Comment (AC2) · 28 Sep 2017

We thank the Reviewers for their valuable and useful comments on this manuscript. Their suggestions will further improve our manuscript and we are certain that we can address the comments in the revised manuscript. Many of the Reviewers' comments highlight the complexity of the problem this paper seeks to address. We hope that this manuscript will add clarity on how storm design and risk might change in the changing world we live in. Please see below our response to each of the reviewers' comment.

[Figure]
We thank the Referee for the valuable and useful comments on this manuscript. We believe that their suggestions will further improve our manuscript.

The paper aims to use observed sub-daily summer (June-October) precipitation intensities from two ARS sites to test for evidence of temporal trends and to build IDF curves using Annual Maximum Series (AMS) and Partial Duration Series (PDS) approaches and a Bayesian method that takes into account the non-stationarity of the time series, using this last approach for a failure analysis addressed to infrastructures that are designed with a stationary approach. The paper is interesting and in the line of some recent literature on the topic. There are few major points to be discussed: 1) The proposed Bayesian approach is interesting but some more details should be provided about the likelihood function and the verification of homoscedastic distribution of residuals. The test is needed if the adopted bayesian likelihood functions is based on such restrictive hypothesis. Otherwise the hypotheses behind the Bayesian approach should be clearly stated (Liuzzo et al. 2017)

The Reviewer raises an excellent point. We will include plots showing the distribution of residuals in the revised version of the manuscript.

2) I understood that Bayesian approach was adopted only for those raingauges for which a local trend was identified (while possible regional crosscorrelation was eliminated by means of RAMK). Bayesian approach is able to generally provide information even if a formal trend cannot be determined providing a sort of "tendency" of the time series to show a trend in the future. This approach can be also replicated in the proposed study with the aim of showing a more general risk analysis

We thank the Reviewer for the comment. We did include all trends, statistically and

non-statistically significant, in the Bayesian analysis. However, we only included in the Figures those trends that were statistically significant. We will incorporate those cases in the revised manuscript.

Lorena Liuzzo, Vincenza Notaro, Gabriele Freni (2017) Uncertainty related to climate change in the assessment of the DDF curve parameters. Environmental Modelling & Software Volume 96, October 2017, Pages 1-13

The text is generally well structured and the figures are all informative. I suggest the publication after the comments are addressed.

Please also note the supplement to this comment:
https://www.hydrol-earth-syst-sci-discuss.net/hess-2017-247/hess-2017-247-AC2-supplement.pdf